# Cloud base height retrieval from multi-angle satellite data

Christoph Böhm[1], Odran Sourdeval[2], Johannes Mülmenstädt[2], Johannes Quaas[2], and Susanne Crewell[1]

[1]Institute for Geophysics and Meteorology, University of Cologne, Cologne, Germany
[2]Institute of Meteorology, University of Leipzig, Leipzig, Germany

*Correspondence to:* Christoph Böhm (c.boehm@uni-koeln.de)

**Abstract.** Clouds are a key modulator of the Earth energy budget at the top of the atmosphere and at the surface. While the cloud top height is operationally retrieved with global coverage, only few methods have been proposed to determine cloud base heights ($z_{base}$) from satellite measurements. This study presents a new approach to retrieve cloud base heights using the Multi-angle Imaging SpectroRadiometer (MISR) on the Terra satellite. It can be applied if some cloud gaps occur within the chosen distance of typically 10 km. The MISR cloud base height (MIBase) algorithm then determines $z_{base}$ from the ensemble of all MISR cloud top heights retrieved at a 1.1-km horizontal resolution in this area. MIBase is first calibrated using one year of ceilometer data from more than 1500 sites within the continental United States of America. The 15th percentile of the cloud top height distribution within a circular area of 10 km radius provides the best agreement with the ground-based data. The thorough evaluation of the MIBase product $z_{base}$ with further ceilometer data yields a correlation coefficient of about 0.66 demonstrating the feasibility of this approach to retrieve $z_{base}$. The impacts of the cloud scene structure and macrophysical cloud properties are discussed. For a three year period, the median $z_{base}$ is generated globally on a $0.25° \times 0.25°$ grid. Even though overcast cloud scenes and high clouds are excluded from the statistics, the median $z_{base}$ retrievals yield plausible results in particular over ocean as well as for seasonal differences. The potential of the full 16 years of MISR data is demonstrated for the southeast Pacific revealing interannual variability in $z_{base}$ in accordance with reanalysis data. The global cloud base data for the three year period (2007–2009) are available at https://doi.org/10.5880/CRC1211DB.19.

## 1 Introduction

As Boucher et al. (2013) state in the IPCC Assessment Report 5, clouds and aerosols continue to contribute the largest uncertainty to estimates and interpretations of the Earth's changing energy budget. To describe the effect of clouds on the radiation energy budget, the geometric thickness, the vertical location of clouds and, therefore, the cloud base height ($z_{base}$) are crucial parameters. Furthermore, long term observations of cloud heights would be beneficial to assess the contribution and the response of clouds to climate change. $z_{base}$ is a key parameter for the radiative energy budget at the Earth surface. $z_{base}$ may also have an impact on ecosystems which are supplied with water by the immersion of clouds (Van Beusekom et al., 2017). Aviation is another field which benefits from information on $z_{base}$.

Various methods to retrieve the $z_{base}$ have been proposed applying different physical concepts, such as active measurements, spectral methods, approaches using an adiabatic cloud model (e.g., Goren et al., 2018), and in-situ measurements.

From the ground, the most accurate and well-established method to derive $z_{base}$ is the backscatter information from a lidar ceilometer, also providing crucial information on visibility for aircraft safety. Thus, ceilometers are employed at airports. Their number has increased in particular in Europe and North America during the past couple of years. A dedicated web page hosted by the Deutscher Wetterdienst shows the distribution of ceilometer stations around the world (http://www.dwd.de/ceilomap).

Radiosondes provide in-situ measurements of thermodynamic variables. Costa-Surós et al. (2014) compare different methods to infer $z_{base}$ from radiosonde data. For the best of these methods, 67 % of the considered profiles agree with the utilized reference data regarding number of cloud layers and height category (distinguished are low, middle and high). Cloud radar transmits microwave radiation to derive vertical profiles of radar reflectivity. However, this signal strongly depends on the particle size. Therefore, the occurrence of a few drizzle drops can mask cloud base. Measurements with radiosondes and cloud

radars are even less common than ceilometers, global coverage cannot be achieved from the ground today.

From space, active measurements are carried out by CALIOP (Cloud Aerosol Lidar with Orthogonal Polarization) on the CALIPSO (Cloud Aerosol Lidar and Infrared Pathfinder Satellite Observations) satellite (Winker et al., 2010). A valid retrieval of the $z_{base}$ can only be ensured if the signal of CALIOP reaches the Earth's surface, which is only possible in case of low optical thickness. Optically thick clouds will lead to attenuation of the signal. The spatial coverage is limited to the narrow laser

beam of CALIOP. The CALIOP cloud base determination has been revisited by Mülmenstädt et al. (2018). They developed an algorithm to extrapolate cloud base retrievals for thin clouds into locations where the CALIOP signal is attenuated within a thicker cloud before it reaches the cloud base.

Passive measurements in the near-infrared exploiting spectral information have been proposed by Ferlay et al. (2010). They suggest an approach to infer the cloud vertical extent from multi-angular POLDER (POLarization and Directionality of the

20 Earth's Reflectances) oxygen A-band measurements. As they point out, the penetration depth of photons into a cloud, and, hence, the height of the reflector, depends on the cloud vertical extent and the viewing geometry. Exploiting the different viewing angles provided by POLDER, Desmons et al. (2013) apply this approach to infer the vertical position of clouds. Their comparison to retrievals from the cloud profiling radar on CloudSat and CALIOP shows that this method works best for liquid clouds over ocean with a retrieval bias of 5 m and a standard deviation of the retrieval differences of 964 m. However, this

approach has not been carried out operationally yet. Moreover, an estimate of the cloud top height is required to retrieve the cloud base height from the cloud vertical extent, which introduces additional uncertainty.

Meerkötter and Zinner (2007) suggest a method to derive $z_{base}$ of convective clouds which are not affected by advective motions. An adiabatic cloud model incorporating measurements of cloud optical depth and effective radius is used to calculate the geometric extent of the cloud from the retrieved cloud top height. By introducing a subadiabatic factor, Merk et al. (2016)

investigate the adiabatic assumption in more detail. By additionally introducing a factor into the calculations, they account for subadiabaticity due to entrainment of dry air through the cloud edges. As a reference, the cloud vertical extent is derived as the difference between $z_{top}$ (radar) and $z_{base}$ (ceilometer) from ground based measurements. The authors conclude that for their two year data set neither the assumption of an adiabatic cloud nor the assumption of a temporally constant subadiabatic factor is fulfilled.

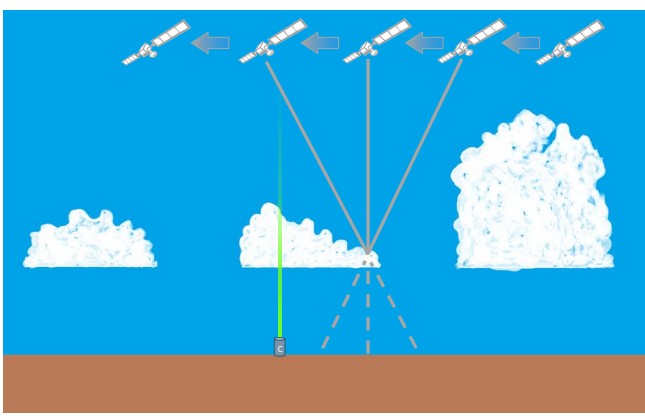

**Figure 1.** Schematic depiction of a cloud field observed from different viewing angles during the satellite overpass. Ceilometers, here represented as a cylindrical box, provide ground-based measurements of cloud base heights which can be used as reference.

Lau et al. (2012) suggest a new approach to determine $z_{\mathrm{base}}$ utilizing the Multi-angle Imaging SpectroRadiometer (MISR) on the Terra satellite. For a preliminary case study, they chose the observations from island Graciosa, Azores, Portugal, for which they compared cloud top height ($z$) retrievals from MISR to collocated and coincidental lidar measurements. Under the assumption that the cloud vertical extent varies horizontally within the cloud, they retrieve $z_{\mathrm{base}}$ by identifying the lowest cloud top height in the height profile provided by MISR. The reference cloud base height ($\hat{z}_{\mathrm{base}}$) is retrieved from the lidar signal by visual inspection of the backscatter coefficient in a time-height cross section over a period of about five hours. They selected 12 cases which show a promising agreement between MISR and lidar retrievals.

We build on the approach proposed by Lau et al. and develop an automatic retrieval method to derive $z_{\mathrm{base}}$ from MISR measurements. Parameters employed in the retrieval scheme are derived from coincident ceilometer measurements over one year in the continental United States of America (USA). The performance of the $z_{\mathrm{base}}$ algorithm is demonstrated by an evaluation with ceilometer over a longer time period and the potential for application on the global scale and for longer time series is explored.

The paper is structured as follows. In Section 2, the utilized data from MISR and from ceilometers are described. Section 3 introduces the new retrieval method along with a case study for illustration. In Section 4, the evaluation of the algorithm against the ceilometer measurements is shown and the effect of the cloud vertical extent on the performance of the algorithm is discussed. Section 5 includes two applications of the algorithm: the median $z_{\mathrm{base}}$ is presented globally for a three-year period, and regionally over the southeast Pacific for a 16-year period. Finally, Section 6 concludes the study.

## 2 Data

### 2.1 MISR cloud product

MISR is carried on board the Terra satellite and provides sun-synchronous (equatorial overpass at around 10:30 local solar time) global products of cloud properties with a 1.1 km horizontal resolution. With an across-track swath width of 380 km,

MISR takes two (poles) to nine (equator) days for repeated observations of the same site. The MISR Level 2TC Cloud Product (MIL2TCSP; Diner, 2012; Moroney and Mueller, 2012; Mueller et al., 2013) is used in this study to provide retrievals of cloud top height $z$ and a stereo-derived cloud mask (SDCM). Three years of global data (2007–2009) are utilized here. The MISR Ancillary Geographic Product (Bull et al., 2011) is additionally used to assign corresponding spatial coordinates and the average scene elevation for each pixel. Here, we give a brief summary on how the operational MISR $z$ product is derived. More in-depth descriptions can be found in Moroney et al. (2002) and in Marchand et al. (2007).

A cloud field is schematically depicted in Fig. 1. MISR hosts cameras providing a total of nine viewing angles. Besides the nadir viewing camera (0°), there are four forward and four aftward viewing cameras set up at 26.1°, 45.6°, 60.0° and 70.5° angles, respectively. During an overpass, each camera of MISR records the reflected radiances at its particular viewing angle. A pattern matching routine which compares the radiances recorded at a wavelength of 670 nm identifies equal cloud features in the images of the different viewing angles. Pixels with the least deviation from each other are matched. This way, a detected cloud feature is observed from multiple satellite positions with its respective time and viewing angle. If at least three images can be attributed to the same cloud feature, the cloud motion vector along with the horizontal and vertical position of the cloud feature can be inferred geometrically. This process is not sensitive to absolute values of the radiances so that this retrieval method is not sensitive to calibration.

The cloud motion vector is determined at a 17.6 km resolution. For each of these coarser grid boxes, the cloud motion vector is then used to determine $z$ at 1.1 km resolution, which is carried out for two camera pairs individually: one pair (FWD) consisting of the nadir and 26.1° forward viewing cameras and the other (AFT) consisting of the nadir and 26.1° aftward viewing cameras. This way, two $z$ values for the same location are available, and the mean of the two values yields the final $z$. In case only one camera pair provides a valid $z$, it is taken as the final $z$ at its specific location. To derive the stereo-derived cloud mask, the two individual $z$ values undergo the following comparison. The retrieval of each camera pair is classified as surface or cloud retrieval according to the threshold height $h_{\mathrm{min}}$ (Equation 1). This is Equation 59 in the Algorithm Theoretical Basis documentation by Mueller et al. (2013), where the threshold height for flat terrain $H_{\mathrm{SDCM}}$ is 560 m, $H$ is the terrain height and $\sigma_{\mathrm{h}}$ is the variance of the the terrain height listed in the Ancillary Geographic Product. Within the MISR Level 2TC Cloud Product, the cloud top height and the stereo-derived cloud mask are also provided without wind correction. Here, we use the the wind corrected data sets.

$$h_{\mathrm{min}} = H_{\mathrm{SDCM}} + H + 2\sigma_{\mathrm{h}} \tag{1}$$

The use of two camera pairs allows attribution of a confidence level to the retrieved $z$. If the mean of the two values is above or below the threshold, the pixel will be classified as cloud or surface, respectively. If only one camera pair provides a valid retrieval, it is tested against the threshold and classified accordingly. In case only one camera pair provides a valid retrieval and in case of two valid retrievals which disagree upon their individual classification, the $z$ retrieval is marked low confidence. If two retrievals are available which agree upon their individual classification, the $z$ retrieval is marked high confidence. Any

other case leads to a non-retrieval. Table 1 summarizes possible combinations of retrievals from the two camera pairs and their corresponding attribution within the stereo-derived cloud mask.

MISR $z$ is given in meters above the World Geodetic System 1984 (WGS 84) surface. To calculate the height above ground level, we subtract the average scene elevation which is provided within the Ancillary Geographic Product for each pixel.

The MISR $z$ product is expected to be superior to $z$ products from other passive instruments. It does not depend on any auxiliary data and it is not sensitive to calibration. Therefore, it is not granted that the application of MIBase to $z$ retrieved by techniques other than the geometric approach would yield similar results.

**Table 1.** Classification scenarios of MISR retrievals. The cloud height obtained using the nadir and the 26.1° forward viewing camera pair (denoted by FWD) and the cloud height obtained using the nadir and the 26.1° aftward viewing camera pair (AFT) are tested against the threshold height $h_{min}$ (Equation 1) individually and then compared to one another to determine the Stereo-Derived Cloud Mask (SDCM) attribute.

| condition | SDCM attribute |
|---|---|
| FWD and AFT above threshold | high confidence cloud |
| FWD and AFT disagree, mean(FWD, AFT) above threshold | low confidence cloud |
| only one camera pair, retrieval above threshold | low confidence cloud |
| FWD and AFT below threshold | high confidence surface |
| FWD and AFT disagree, mean (FWD, AFT) below threshold | low confidence surface |
| only one camera pair, retrieval below threshold | low confidence surface |

## 2.2   METAR data

Aerodrome routine meteorological reports (METAR) (WMO; World Meteorological Organization, 2013) contain weather ob-
servations at airports worldwide, including measurements of $z_{base}$. METARs from airports from the continental USA provide $z_{base}$ determined by the Automated Surface Observing System (ASOS; National Oceanic and Atmospheric Administration, Department of Defense, Federal Aviation Administration, and United States Navy, 1998). ASOS utilizes lidar ceilometers which operate at a wavelength of 0.9 μm and have a vertical range of 12000 ft ($\approx 3700$ m). Cloud base heights are routinely retrieved by evaluating the vertical gradient of the detected backscatter profile with a temporal resolution of 30 seconds. These
individual retrievals are stored in different bins by rounding to the nearest 100 ft ($\approx 30$ m) for heights between the surface and 5000 ft ($\approx 1500$ m); to the nearest 200 ft ($\approx 60$ m) for heights between 5000 ft ($\approx 1500$ m) and 10000 ft ($\approx 3000$ m); and to the nearest 500 ft ($\approx 150$ m) for heights above 10000 ft ($\approx 3000$ m). If there are more than five bins filled with measurements during a 30 minute period, the cloud heights are clustered into layers until only five cluster remain. Finally, all cluster heights are rounded according to the rules given in Tab. 2. The lowest three layers are passed on to the METAR message.
We extract the ceilometer cloud base height $\hat{z}_{base}$ from METAR data for a total of 1510 ceilometer sites around the continental USA to benefit from the homogeneity of the automated measurements and the standardized reporting range. $\hat{z}_{base}$ serves as

**Table 2.** The ceilometer $\hat{z}_{base}$ retrievals are rounded to different values depending on their height window according to ASOS User Guide (National Oceanic and Atmospheric Administration, Department of Defense, Federal Aviation Administration, and United States Navy, 1998). The values are originally given in feet and are converted to meters here.

| height [ft] | rounded to nearest value [ft] | rounded to nearest value [m] |
|:---:|:---:|:---:|
| < 5000 | 100 | 30.5 |
| 5000 to 10000 | 500 | 152 |
| > 10000 | 1000 | 305 |

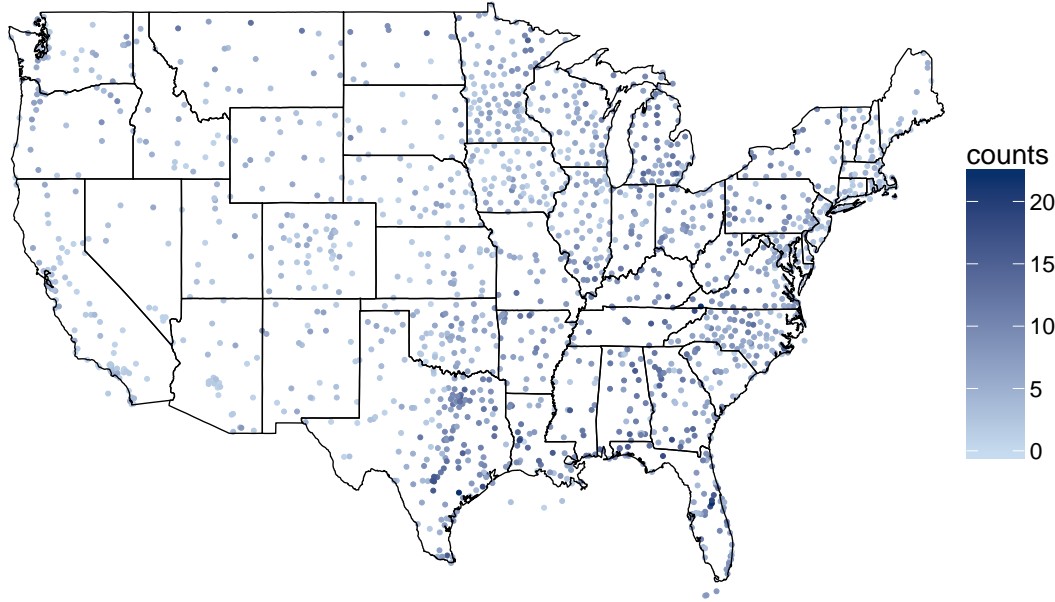

**Figure 2.** Locations of ceilometer stations utilized in this study across the continental USA. Data from these stations for the years 2008 and 2007 are used for the calibration of the $z_{base}$ retrieval algorithm and a subsequent evaluation, respectively. Blue shading indicates the number of valid coincidental retrievals from MISR and ceilometers which have been utilized for the calibration (year 2008) and are within the constraints described in the text.

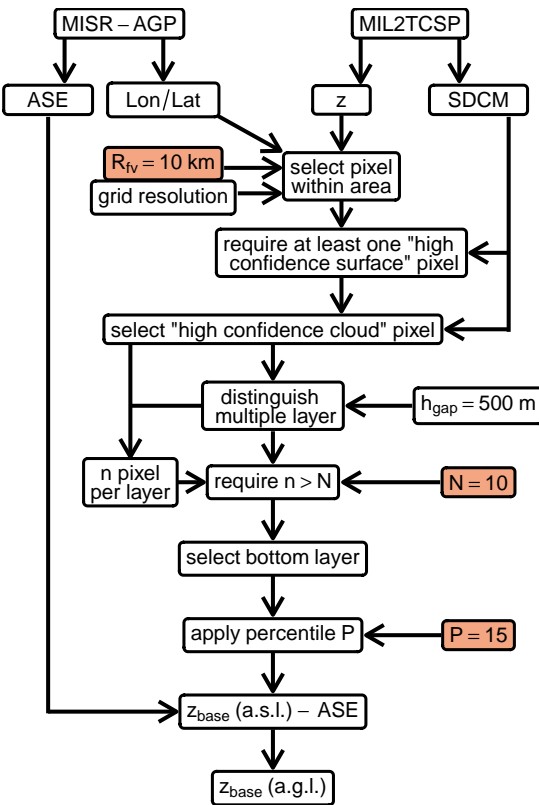

**Figure 3.** Flow chart of the $z_{\text{base}}$ retrieval algorithm. MISR's MIL2TCSP cloud product provides $z$ and the Stereo-Derived Cloud Mask (SDCM). MISR's Ancillary Geographic Product (MISR-AGP) provides the Average Scene Elevation (ASE) and the longitude and latitude coordinates for each pixel. Starting from these products, the depicted processing steps are undergone to derive $z_{\text{base}}$. The parameters which have been optimized during the calibration are highlighted in orange.

reference data to which the $z_{\text{base}}$ derived from the satellite cloud heights is compared. First, METAR data from 2008 are used to estimate parameters used in the $z_{\text{base}}$ retrieval algorithm to create the MISR Cloud Base height algorithm (MIBase). Second, to validate the "tuned" algorithm, METAR data from 2007 are applied for a statistically independent comparison. For a total of 1510 ceilometer stations, collocated and coincidental satellite-based $z_{\text{base}}$ retrievals could be found (see below for exact definition). A distribution of the locations can be seen in Fig. 2.

## 3   Cloud Base height retrieval

The MISR Cloud Base height retrieval (MIBase) algorithm, which derives $z_{\text{base}}$ from the MISR $z$ product, is developed and calibrated with collocated METAR data for defining the involved parameters and preconditions. The first section of this chapter introduces the retrieval principle on the basis of a case study. By comparison with METAR ceilometer measurements from

2008, parameters used within MIBase are estimated, namely the radius $R_c$ of the MIBase retrieval cell, the minimum number of valid cloud pixel $N$ and the percentile $P$ of the $z$ distribution.

## 3.1 Method

We assume that the information on the $z_{base}$ is included in the distribution of the $z$ retrievals from the MISR cloud product for a specific area of limited size. This assumption is valid in a cloud scene with a homogeneous $z_{base}$ and a heterogeneous $z$ similar to the one schematically depicted in Fig. 1. Especially at the edge of a cloud where the cloud is thinner, $z$ can serve as a proxy for $z_{base}$. To ensure that the thinner edge of the cloud is within the observed MIBase retrieval cell, the considered area needs to be large enough and the cloud field needs to be broken. The inherent assumption of a homogeneous $z_{base}$ over a certain area presupposes a horizontally constant lifting condensation level. This is in particular given in a well mixed boundary layer or a homogeneous air mass away from the proximity of a frontal zone, where advective motion could introduce temperature or humidity gradients across the horizontal plane.

In order to derive $z_{base}$ from the $z$ product, the following steps, which are outlined in Fig. 3, are undertaken. First, a retrieval cell has to be defined. For the comparison to the ceilometer measurements, we consider a circular area with the radius $R_c$ around its midpoint at a ceilometer station. In order to estimate the magnitude of $R_c$, we consider the following: METAR $\hat{z}_{base}$ retrievals are representative for a time window of 30 minutes. Within this time window and at a typical wind speed of approximately $10\,\text{ms}^{-1}$, a cloud would shift its position about $20\,\text{km}$ in the wind direction. Therefore, the magnitude of $R_c$ should be on the order of kilometers. The impact of $R_c$ on the retrieved $z_{base}$ and, therefore, the deviation from the ceilometer $\hat{z}_{base}$ is discussed below. When we apply the algorithm to retrieve a global estimate of $z_{base}$, we use a regular lat-lon grid of $0.25°$ (cf. Section 5). This grid size corresponds to a meridional length of the grid boxes of about $28\,\text{km}$ and a zonal length ranging between $25\,\text{km}$ ($25°\text{N}$) and $18\,\text{km}$ ($50°\text{N}$), taking the continental U.S.A. as an example. A greater MIBase cell increases the chance of seeing the thinner part of the cloud. This could lead to a more realistic $z_{base}$ retrieval. In turn, for a smaller MIBase cell the assumption of a homogeneous $z_{base}$ is more realistic.

For each grid cell or circular MIBase cell, the enclosed $z$ retrievals from the MISR cloud product are processed further. MIBase only selects those $z$ retrievals which are marked high confidence cloud (hcc) according to the stereo-derived cloud mask. A consideration of retrievals marked low confidence cloud has shown a decrease of the correlation with the ceilometer $\hat{z}_{base}$. An example of a cloud field with $z$ retrievals and the corresponding stereo-derived cloud mask for 21 August 2015 at the International Airport of Atlanta, Georgia, USA, is presented in Fig. 4 (left, middle).

For some scenes, the distribution of $z$ reveals extended height ranges with no $z$ retrievals between two or more local maxima. Such cases suggest multi-layer cloud scenes if the apparent gap between adjacent $z$ retrievals is of sufficient size. If such a gap $h_{gap}$ is greater than $500\,\text{m}$, the algorithm distinguishes between the cloud layer above and below the gap (cf. Fig. 4 (right) for the aforementioned example). The value for this threshold has been chosen to be close to the specified accuracy of MISR ($560\,\text{m}$). By evaluating different vertical cloud layers individually, a $z_{base}$ retrieval for each layer can be derived. Since for most applications the lowest $z_{base}$ is of interest, the lowest detected cloud layer is processed here. For the comparison with $\hat{z}_{base}$, we restrict ourselves to scenes for which MISR detects only one cloud layer.

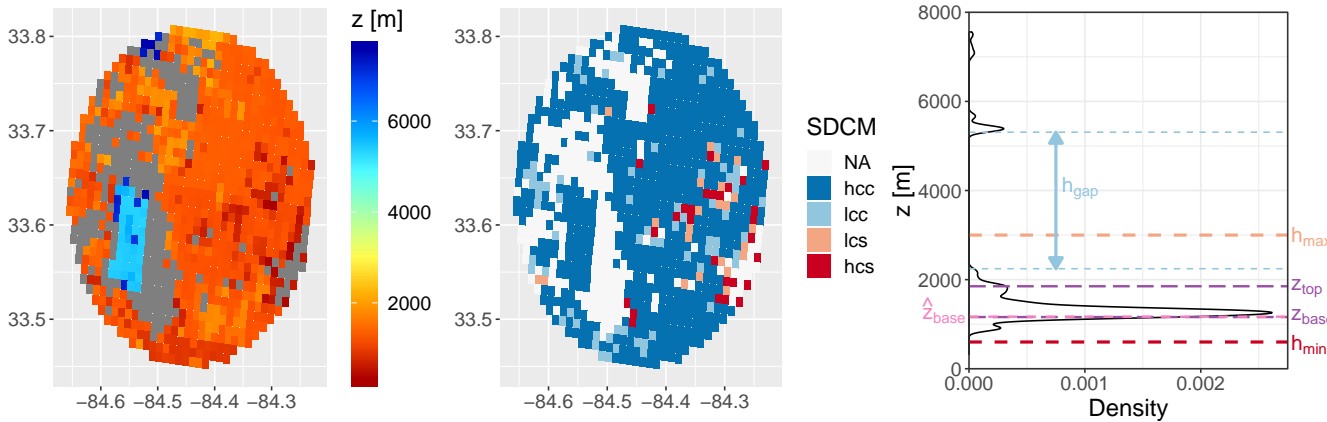

**Figure 4.** MISR observations within a 20 km radius within the vicinity of Atlanta, Georgia, USA (ICAO:KATL) on 21 August 2015 at around 16:30 UTC. Left: $z$. Middle: Corresponding Stereo Derived Cloud Mask (SDCM) distinguishing non-retrievals (NA), high confidence cloud (hcc), low confidence cloud (lcc), low confidence surface (lcs) and high confidence surface (hcs). Right: Density of $z$ measurements with illustration of certain parameters: height between two layers ($h_{gap}$) which is the height difference between the highest retrieval of the bottom layer and the lowest retrieval of the next higher layer (dashed blue lines), upper cut-off height (dashed orange) for $z_{base}$ retrievals ($h_{max}$) which is based on the ceilometer granularity, lower cut-off height (dashed red), which is based on the MISR threshold height to distinguish between cloud and surface retrieval ($h_{min}$), and the ceilometer retrieval $\hat{z}_{base}$ from 16:52 UTC (dashed pink). $z_{top}$ and $z_{base}$ (dashed purple) are inferred by applying the 15th and 95th percentile to the distribution of $z$ of the lowest cloud layer, respectively. Heights are above sea level.

The occurrence of a broken cloud field is a basic requirement of MIBase. Therefore, at least one $z$ retrieval marked high confidence surface needs to be within the MIBase cell. A complete cloud cover or a high rate of non-retrievals can prevent this criterion from being met. Both scenarios suggest doubtful $z_{base}$ retrievals. Hence, they are not considered.

For each grid cell or circular cell surrounding the ceilometer station, $z_{base}$ is diagnosed from the height distribution of $z$ using
a certain percentile $P$. In principle, $P$ should be as low as possible. However, as a certain measurement noise is expected and a robust result should be achieved, a choice substantially larger than zero is necessary. Another parameter which describes the distribution of $z$ for each scene is the number of valid $z$ retrievals marked high confidence cloud $n$. A higher $n$ implies a higher observed cloud cover within the MIBase cell. In order to take a meaningful percentile of the $z$ distribution, a minimum $n > N$ is required. A cloud which is horizontally more extended (higher cloud cover) is more likely to pass over the ceilometer, so that
there is a higher chance that both instruments observe the same cloud. Therefore, the deviation of $z_{base}$ from $\hat{z}_{base}$ is expected to decrease for a higher $n$. The impact of the threshold for $N$ is studied later on.

For certain applications, the cloud vertical extent $\Delta z$ might be of interest. Therefore, an estimate of the cloud top height $z_{top}$ is required. In principle, $P = 100$ should yield the highest point of the cloud. However, analogously to the retrieval of $z_{base}$, a certain measurement noise is expected, so that $P$ is not chosen to be the extreme value. Without further validation, we apply the
95th percentile rather than the median, as we do not want a height which might be representative for the whole area, but rather

an estimate of the highest top of the cloud especially for a heterogeneous cloud top height to estimate $\Delta z$ at its most extensive point.

## 3.2 Case study

One of the utilised ceilometer stations is located at the Hartsfield–Jackson Atlanta International Airport. To illustrate the functionality of the presented algorithm, we investigate a particular MISR overpass over this station on 21 August 2015 at around 16:30 UTC. Figure 4 shows the $z$ retrievals for all pixels which are within the circular MIBase cell defined by $R_c$. Here, we exemplarily use $R_c = 20$ km with its midpoint at the ceilometer station. $z$ is given above the WGS 84 surface, which is approximately equal to sea level. The spatial distribution shows a low cloud layer with $z$ between 800 m and 2000 m, which covers most of the area. Another cloud layer appears between 5 km and 6 km. Some pixels with heights above 7 km indicate the presence of a third layer (Fig. 4, left). For a few pixels, MISR was not able to determine $z$. This might be due to the viewing geometry. A retrieval requires valid images from two different cameras, one camera viewing nadir and the other viewing at a 26.1° angle. In the case studied here, the most missing retrievals are closely attached to high clouds which might lead to shading effects (Fig. 4, middle).

The density of the $z$ distribution shows the aforementioned three cloud layers. They are distinguished according to the threshold value for $h_{gap}$ (Fig. 4, right) as illustrated for the bottom and middle layer. For the bottom layer, which is selected for further processing, the number of $z$ retrievals marked high confidence cloud is determined to be $n = 621$. This number is well above the threshold $N$ which is defined later. $z_{base}$ is then calculated using $P = 15$ as the preliminary percentile of the $z$ distribution. This yields $z_{base} \approx 1160$ m above the WGS 84 surface. The mean average scene elevation for the given area is subtracted from the retrieval to obtain $z_{base} \approx 927$ m above ground level. The closest METAR report for this day is from 16:52 UTC. Three heights were reported at 2800 ft ($\approx 853$ m), 7500 ft ($\approx 2286$ m) and 23000 ft ($\approx 7010$ m) above ground level. By adding the station elevation (315 m), the corresponding height above sea level is obtained. This yields $\hat{z}_{base} \approx 1168$ m and is denoted in Fig 4 (right). In conclusion, using the preliminary values for $P$ the $z_{base}$ retrieval from MISR is about 927 m above ground level which is 74 m higher than the ceilometer retrieval ($\hat{z}_{base} = (853 \pm 15)$ m). The given uncertainty solely represents the resolution of the METAR reports (Tab. 2). Note that the third layer detected around 7000 m by MISR has also been detected by the ceilometer.

## 3.3 Parameter optimization

For each considered ceilometer station (Fig. 2), collocated and coincidental MISR overpasses from the year 2008 are identified. The algorithm is then applied as described in the case study (Sec. 3.2 to retrieve $z_{base}$. All pairs of MIBase $z_{base}$ and ceilometer $\hat{z}_{base}$ are evaluated to investigate the influence of $R_c$, $N$ and $P$ on the performance of the $z_{base}$ retrieval algorithm and to estimate the most suitable values. For this purpose, the following statistical measures are considered: the slope and intercept of a linear regression, which are ideally 1 and 0, respectively; the Pearson correlation coefficient $r$ (ideally unity); the root mean square

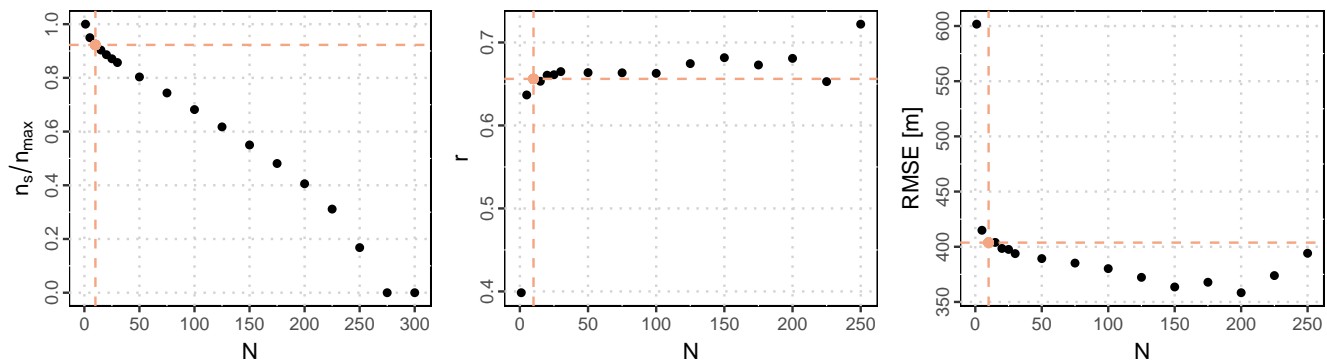

**Figure 5.** Evaluation of minimum number of valid pixels $N$ within a cloud layer detected by MISR for the year 2008. Left: The normalized number of events $\frac{n_s}{n_{max}}$ for which $z_{base}$ and $\hat{z}_{base}$ could both be retrieved. $n_{max}$ is the maximum number of events, which is found for $N = 1$. Middle: The linear correlation coefficient $r$ between $z_{base}$ and $\hat{z}_{base}$. Right: The RMSE between $z_{base}$ and $\hat{z}_{base}$. MISR $z_{base}$ is retrieved using the 15th percentile of the $z$ distribution for a 10 km radius around the individual ceilometer measurements. The chosen value for $N$ is highlighted in orange. For further details see text.

error (RMSE) $E$ defined as

$$E = \sqrt{\frac{1}{n} \sum_{i=1}^{n} \left( z_{base,i} - \hat{z}_{base,i} \right)^2}; \tag{2}$$

and the retrieval bias $B$ defined as

$$B = \frac{1}{n} \sum_{i=1}^{n} \left( z_{base,i} - \hat{z}_{base,i} \right). \tag{3}$$

**Table 3.** Slope, intercept, correlation coefficient $r$, RMSE $E$, bias $B$ and number of samples $n_s$ resulting from comparing $z_{base}$ and $\hat{z}_{base}$ retrievals for different radii of the MISR circular area around the ceilometer stations. These values are obtained for the year 2008 applying a required minimum number of cloud pixels of $N = 10$ and the 15th percentile to the $z$ distribution.

| $R_c$ | slope | intercept | $r$ | $E$ | $B$ | $n_s$ |
|---|---|---|---|---|---|---|
| [km] | | [m] | | [m] | [m] | |
| 5 | 0.65 | 371 | 0.66 | 392 | -71 | 3059 |
| 10 | 0.62 | 412 | 0.66 | 404 | -75 | 5120 |
| 15 | 0.60 | 433 | 0.65 | 413 | -77 | 6140 |
| 20 | 0.58 | 464 | 0.63 | 423 | -74 | 6895 |
| 30 | 0.54 | 515 | 0.60 | 437 | -71 | 7772 |

5    MISR can only detect clouds above the threshold height according to Equation 1. To prevent this obvious limitation from introducing a bias into the statistics, we only consider cloud scenes for which the ceilometer retrieval is above $h_{min}$. In addition,

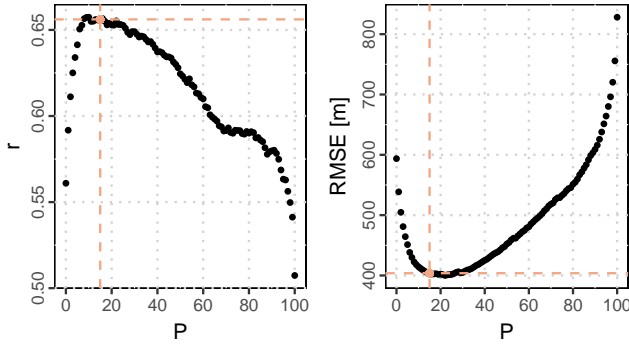

**Figure 6.** Evaluation of the percentile $P$ which is applied to retrieve $z_{base}$ from the distribution of $z$ for the year 2008 with $N = 10$ and $R_c$ = 10km. Left: The linear correlation coefficient $r$ between $z_{base}$ and $\hat{z}_{base}$. Right: The RMSE between $z_{base}$ and $\hat{z}_{base}$. The chosen value for $P$ is highlighted in orange.

only $z_{base}$ retrievals below a maximum height $h_{max}$ of 3000 m are considered to focus on a cloud range for which the ceilometer retrievals are more finely granulated (below 10000 ft according to Tab. 2).

First, we investigate the influence of the size of the MIBase cell on the comparison of MIBase and ceilometer retrievals. For this purpose, $R_c$ is varied between 5 and 30 km while the other parameters are set to the preliminary values $P = 15$ and

$N = 10$. With a decreased $R_c$, the correlation between $z_{base}$ and $\hat{z}_{base}$ increases and $E$ decreases (Tab. 3). This is to be expected as the representativity should increase. However, for a lower $R_c$, the retrieval algorithm encounters more situations where at least one of the requirements (at least one high confidence surface pixel is visible and at least 10 valid cloud pixel per layer) cannot be fulfilled, as the decrease in the total number of retrievals indicates. The better agreement between $z_{base}$ and $\hat{z}_{base}$ for lower $R_c$ might be due to a relatively larger overlap of the measurement sampling areas of the two instruments and to a better

fulfilment of the assumption of a homogeneous $z_{base}$ over smaller areas. For further evaluation, a radius of 10km is chosen as a compromise between a good agreement in terms of $r$ and $E$ and without having to discard too many retrieval scenes.

Second, the effect of the minimum number of of valid $z_{base}$ retrievals is studied which strongly limits the number of samples for the comparison (Fig. 5). With increasing $N$, initially a slight increase to $N = 10$ improves the correlation between $z_{base}$ and $\hat{z}_{base}$ and $E$ significantly to a correlation coefficient of about 0.66. A further increase only yields slight improvement of

the correlation and $E$. This slight increase can be explained by the elimination of more complex scenes from the comparison. However, for a higher $N$ the trade off is a lower total number of $z_{base}$ retrievals. For instance, for $N = 50$ only 80 % of possible retrievals yield a valid $z_{base}$ (Fig. 5, left). Therefore, we select $N = 10$.

Finally, we consider the percentile threshold used to diagnose $z_{base}$ from the $z$ distribution. Figure 6 shows an evaluation of different percentiles which are applied to derive $z_{base}$. Percentiles between the 10th and the 15th give the best correlation. The

lowest $E$ is achieved for percentiles between the 15th and the 25th. Therefore, $P = 15$ is chosen for further processing. The fact that very clear and localised minima (maxima) for $E$ ($r$) are found supports the hypothesis that the $z$ distribution contains information on $z_{base}$.

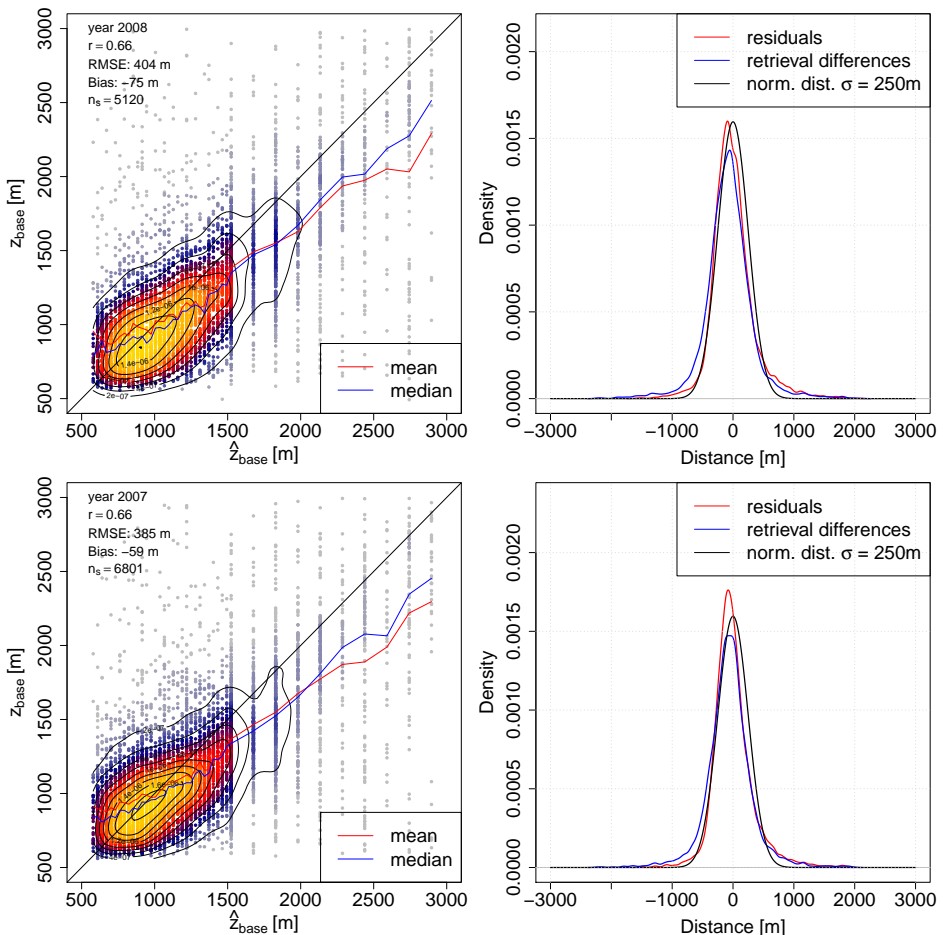

**Figure 7.** Left: Joint density of $z_\text{base}$ and $\hat{z}_\text{base}$ for the year 2008 (top) which is used to estimate parameters of the algorithm and for the year 2007 (bottom) which is used to validate the stability of the algorithm with the estimated parameters. The value of the normalized density is indicated by color (maximum values in light yellow) and contour lines with corresponding values on them (linear scale). For each ceilometer height bin the mean (red) and median (blue) of the MISR $z_\text{base}$ is shown. Right: Probability density functions of the residuals after a linear fit (red), the retrieval differences (blue) and a normal distribution with a standard deviation of 250 m (black).

In summary, the comparison yields the estimated parameters $R_c = 10$ km, the minimum number $N = 10$ and the percentile $P = 15$. While the latter two are kept fixed in MIBase, $R_c$ is optimised for the intercomparison with point data, i.e. ceilometer measurements. The algorithm can also be applied to larger grids. However, no data for validating extended areas are available.

**Table 4.** Slope, intercept, correlation coefficient $r$, RMSE $E$, bias $B$ and number of retrievals $n_s$ resulting from a comparison of $z_{base}$ and $\hat{z}_{base}$ for data obtained 2008 (calibration) and 2007 (validation). These values are obtained with $N = 10$ and $P = 15$.

| data | pixel/grid definition | slope | intercept [m] | $r$ | E [m] | B [m] | $n_s$ |
|------|------|------|------|------|------|------|------|
| 2008 | $R_c = 10$ km | 0.62 | 412 | 0.66 | 404 | -75 | 5120 |
| 2007 | $R_c = 10$ km | 0.61 | 419 | 0.66 | 385 | -59 | 6801 |
| 2007 | $0.25° \times 0.25°$ | 0.58 | 455 | 0.64 | 398 | -60 | 7970 |
| 2007 | $0.75° \times 0.75°$ | 0.49 | 579 | 0.55 | 446 | -56 | 10474 |

## 3.4 Scene limitations

This section investigates the applicability of MIBase by quantifying the amount of cases for which the concurrent conditions allow the successful derivation of a $z_{base}$ retrieval. First, we filter for cases which fulfill the following two conditions: i) The number of valid $z$ retrievals within the MIBase cell $N_{val}$ must be $> 0$ and ii) METAR data must be available for the calibration and validation. These requirements are fulfilled for about two thirds of a all considered MISR overpasses over the ceilometer sites (Table 5). Furthermore, there are two main conditions which prevent the derivation of a $z_{base}$ retrieval. These are namely apparent clear sky conditions and apparent overcast which is only a limitation for MIBase. Here, we use the phrases "apparent clear sky" and "apparent overcast" rather than "clear sky" and "overcast", respectively, to account for the fact that this attribution is based on instrumental indications rather than known actual sky condition.

For METAR, apparent clear sky is indicated if a METAR message is available, but does not provide a valid retrieval. Note that in case the lowest cloud is above the METAR reporting range (typically 3700 m), it is possible that no retrieval is issued. Here, such cases would also be attributed apparent clear sky.

For MIBase, we attribute apparent clear sky to the following configuration of the SDCM: MISR sees the surface with high confidence ($N_{HCS} > 0$), and has no high confidence cloud in the view ($N_{HCC} = 0$). This does not have to be an actual clear sky case since it could include low confidence surface or low confidence cloud retrievals for which the declaration is less certain. In case of invalid $z$ retrievals, it is also uncertain whether clouds are present or not.

Out of all MISR apparent clear sky cases, 87 % are also classified as clear sky by METAR while the remaining 13 % yield a METAR cloud height retrieval. Mismatches in attributing apparent clear sky cases are due to METAR retrievals below the threshold height $h_{min}$ (17 %) and other reasons, such as the temporal offset between MISR and METAR measurement. The METAR reports comprise retrievals over a 30 minute period. During this time, cloud formation and cloud dissipation can alter the cloud scene and cause mismatches between MISR and METAR retrievals.

**Table 5.** Number of cases for different conditions of the cloud field observed by MISR and reported in METAR messages for the considered METAR sites. The number of $z$ retrievals labeled "high confidence cloud" ($N_{HCC}$) or "high confidence surface" ($N_{HCS}$) according to MISR's stereo-derived cloud mask is used to characterize the cloud field. The size of the scene is defined by $R_c = 10$ km. * indicates apparent conditions. See text for details, including the meaning of boldface font.

| description of the situation | | 2008 | 2008 | 2007 | 2007 |
|---|---|---|---|---|---|
| MISR | METAR | | [%] | | [%] |
| overpasses over METAR sites | | 80454 | 154.1 | 89782 | 145.9 |
| valid $z$ retrievals | message available | **52215** | **100.0** | **61531** | **100.0** |
| $N_{HCC}=0; N_{HCS}>0$ (clear sky*) | | 19507 | 37.4 | 20300 | 33.0 |
| | clear sky* | 26983 | 51.7 | 30037 | 48.8 |
| clear sky* | clear sky* | **16982** | **32.5** | **17374** | **28.2** |
| $N_{HCC}=0; N_{HCS}>0$ (clear sky*) | $\hat{z}_{base}$ retrieval | **2525** | **4.8** | **2926** | **4.8** |
| $N_{HCC}=0; N_{HCS}>0$ (clear sky*) | $\hat{z}_{base}>h_{min}$ | 2106 | 4.0 | 2520 | 4.1 |
| $N_{HCC}>0; N_{HCS}>0$ | clear sky* | **6800** | **13.0** | **8511** | **13.8** |
| $N_{HCC}>0; N_{HCS}=0$ (overcast*) | | **15945** | **30.5** | **19725** | **32.1** |
| $N_{HCC}>0; N_{HCS}=0$ (overcast*) | $\hat{z}_{base}$ retrieval | 12769 | 24.5 | 15600 | 25.4 |
| $N_{HCC}>0; N_{HCS}=0$ (overcast*) | clear sky* | 3176 | 6.1 | 4125 | 6.7 |
| $N_{HCC}=0; N_{HCS}=0$ | | **51** | **0.1** | **51** | **0.1** |
| $N_{HCC}>0; N_{HCS}>0$ | $\hat{z}_{base}$ retrieval | **9912** | **19.0** | **12944** | **21.0** |
| $N_{HCC}\geq N=10; N_{HCS}>0$ | $\hat{z}_{base}$ retrieval | 8603 | 16.5 | 11387 | 18.5 |
| $z_{base}$ retrieval | $\hat{z}_{base}$ retrieval | 8535 | 16.3 | 11319 | 18.4 |
| $z_{base}$ retrieval; single layer | $\hat{z}_{base}$ retrieval | 7863 | 15.1 | 10251 | 16.7 |
| $z_{base}<h_{max}=3$ km; single layer | $\hat{z}_{base}$ retrieval | 7206 | 13.8 | 9407 | 15.3 |
| $z_{base}<h_{max}$; single layer | $\hat{z}_{base}<h_{max}$ | 7043 | 13.5 | 9227 | 15.0 |
| $z_{base}<h_{max}$; single layer | $h_{min}<\hat{z}_{base}<h_{max}$ | **5120** | **9.8** | **6801** | **11.1** |

Furthermore, for MIBase, we attribute apparent overcast to the following configuration of the SDCM: MISR observes a cloud with high confidence ($N_{HCC}>0$) and does not observe any surface retrievals with high confidence ($N_{HCS}=0$). Again, the scene could include invalid retrievals, or retrievals of low confidence. In about 20 % of all the MISR apparent overcast cases, the corresponding METAR report yields an apparent clear sky case. These could be cases where the cloud cover is mainly above the reporting range of the ceilometer.

Out of all cases with valid $z$ retrievals within the MIBase cell ($N_{val}>0$) and a corresponding METAR retrieval, 19 % are processed further. The main reasons why cases are excluded are apparent clear sky scenes for MISR (37.4 %), apparent overcast for MISR (30.5 %) and apparent clear sky for METAR when valid $z$ retrievals are within the MIBase cell (13 %). Additional requirements, such as the minimum number of $z$ retrievals marked high confidence cloud ($N_{HCC}>N$), single layer situations, $z_{base}$ and $\hat{z}_{base}$ retrievals below $h_{max}$ and METAR retrievals above the MISR threshold height ($\hat{z}_{base}>h_{min}$), lead to a further

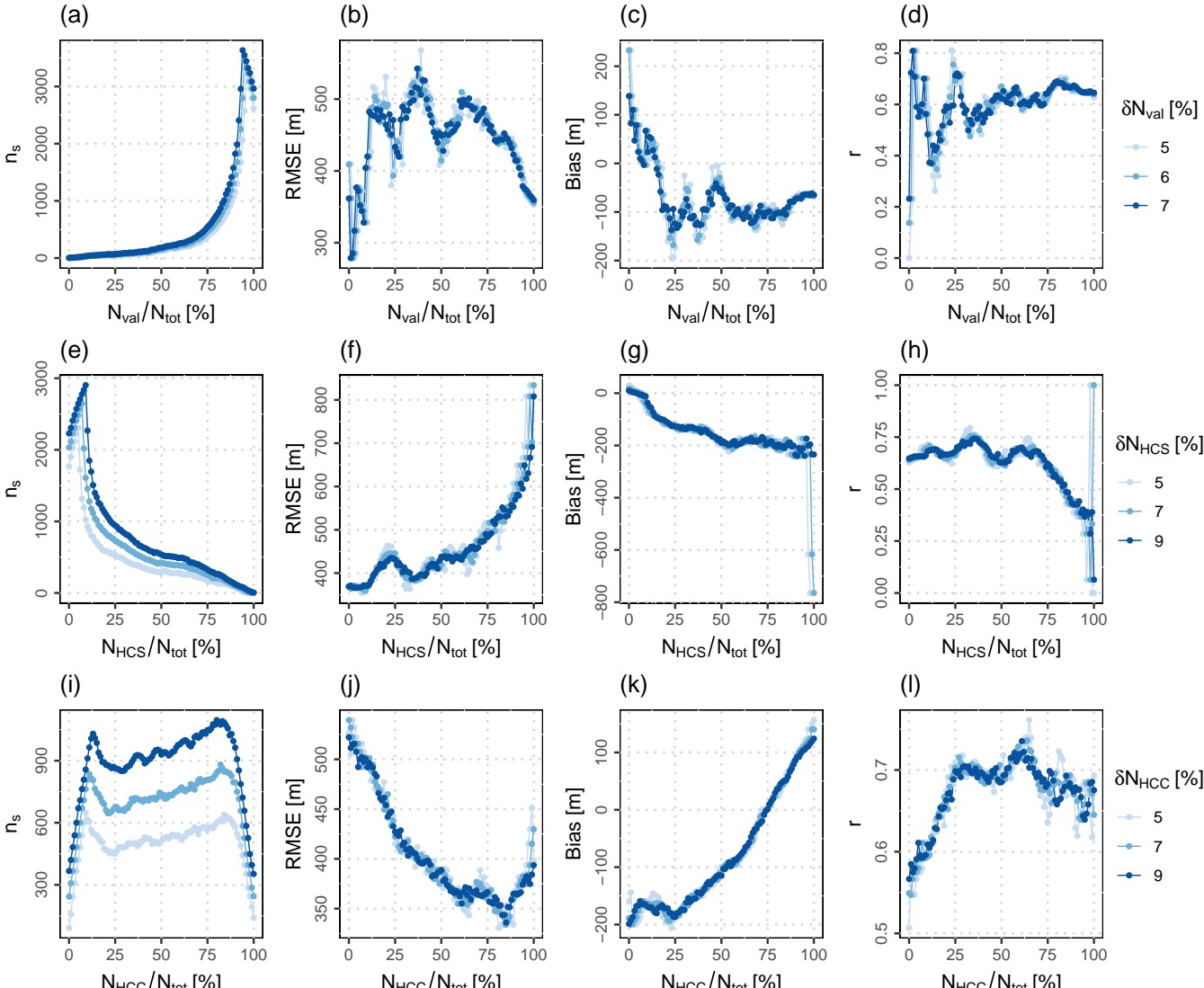

**Figure 8.** From left to right: number of samples $n_s$, RMSE, bias and correlation coefficient $r$ for the comparison of MIBase and ceilometer retrievals as a function of the number of valid $z$ retrievals $N_{val}$ (top row), the number of retrievals marked high confidence surface $N_{HCS}$ (middle row) and the number of retrievals marked high confidence cloud $N_{HCC}$ (bottom row). Each data point is calculated for a sub sample which includes only $N_{val} \pm \delta N_{val}$, $N_{HCS} \pm \delta N_{HCS}$ and $N_{HCC} \pm \delta N_{HCC}$, respectively. The various widths of the considered $N_{val}$ and $N_{HCC}$ windows are indicated by the blue shading. All values are normalized by the total number of pixels within the MIBase cell $N_{tot}$. Data are for the year 2008 with $R_c = 10\,km$, $P = 15$ and $N = 10$.

reduction of the number of cases which are used to derive the statistics. Further numbers for specific cases are presented in Table 5.

## 4 MIBase Evaluation

With the parameters $R_c = 10\,\text{km}$, $N = 10$ and $P = 15$ derived in the previous section, MIBase is applied to MISR retrievals which are coincident with ceilometer retrievals from the year 2007. These data have not been used for calibration. The joint density of $z_{\text{base}}$ retrieved from MISR and ceilometer is shown in Fig. 7. For lower $z_{\text{base}}$, MISR yields higher heights than the

ceilometers. This can possibly be attributed to the threshold height (Equation 1) constraining $z_{\text{base}}$ retrievals at the lower end of the height distribution. For $z_{\text{base}}$ greater than 1000 m, mean and median MISR heights are lower than the ceilometer. Overall, the bias $B$ is slightly negative (about 60 m; cf. Tab. 4) and the density of the retrieval differences is shifted slightly towards negative values (Fig. 7, right). Thus, MISR $z_{\text{base}}$ retrievals are generally lower than the ceilometer retrievals. This could be due to the different sample volumes. On the one hand, the ceilometer only records point measurements over a period of time, so that

the measured sample of the cloud depends on the velocity of the wind. On the other hand, MISR observes the entire circular area defined by $R_c$ around the ceilometer location. Chances are that MISR can observe a cloud with a lower base which does not pass over the ceilometer.

The joint density and the density of the retrieval differences appear similar for both the 2007 and the 2008 data sets (Fig. 7). Slope, intercept, $r^2$, $E$, and $B$ resulting from the $z_{\text{base}}$ retrieval comparisons for the year 2008 (calibration) and the year 2007

(validation) appear very similar, demonstrating the stability of the algorithm with the chosen parameters (Tab. 4) to interannual variability in cloud properties. Changing the MIBase cell to a $0.25° \times 0.25°$ latitude–longitude grid results in a slightly lower correlation coefficient accompanied by a higher $E$. An even coarser grid size of $0.75° \times 0.75°$, which is applied later for a comparison with ERA-Interim cloud heights, results in an even lower correlation and higher $E$. A decreasing agreement between $z_{\text{base}}$ and $\hat{z}_{\text{base}}$ for a larger MIBase cell has already been described when studying the influence of $R_c$ (see discussion

in Section 3.3).

### 4.1 Scene structure influence

To estimate the influence of the the scene structure on the performance of MIBase, we further exploit the MISR cloud top height product and the MISR Ancillary Geographic Product to investigate characteristics of the terrain height and the cloud field.

To derive a quantity to estimate the variability of the terrain height, we calculate the standard deviation of the average scene elevation, which is provided by the ancillary product at 1.1 km resolution. For each METAR site, the standard deviation is calculated for an area defined by different $R_c$ (5 km, 10 km, 15 km, 20 km and 30 km). Typical standard deviations range around a few tens of meters with overall higher standard deviations for greater $R_c$ (Fig. S1 a). When METAR sites with a higher standard deviation of the average scene elevation are excluded from the comparison of MIBase and METAR cloud base height

retrievals, the RMSE decreases slightly, the bias slightly increases (towards 0), while the correlation is hardly affected (Fig. S1 b,c,d). Thus, the variability of the terrain height has a very small effect on the accuracy of the MIBase algorithm, with a slightly better performance over more homogeneous terrain.

To further investigate the performance of the MIBase algorithm as a function of parameters related to cloud types, we determine RMSE, bias, and the correlation coefficient as a function of $z_{\text{top}}$ and the cloud vertical extent $\Delta z$ (Fig. S2). The best correlation is obtained for cloud vertical extents up to 1000 m. The RMSE is also smaller for lower $\Delta z$ and for lower $z_{\text{top}}$. However, the RMSE increases with decreasing $z_{\text{top}}$ below about 1000 m. We conclude that MIBase performs best for shallow low clouds. However, further analyses are necessary to increase the sample size of thicker clouds and to include more medium high and high clouds for a more robust analysis of such cloud types. Furthermore, the increased RMSE for very low $z_{\text{top}}$ indicates that, for very shallow low clouds in the proximity of the threshold height, MIBase retrievals do not agree as well with the METAR retrievals. This might be due to cases for which MIBase detects a shallow low cloud with $z_{\text{base}}$ and $z_{\text{top}}$ close the $h_{\text{min}}$ when, in fact, the actual cloud base is below $h_{\text{min}}$. MIBase would miss this actual cloud base height because the retrievals below $h_{\text{min}}$ would not be marked high confidence cloud. For that matter, we require that the ceilometer retrieval is above the threshold height ($\hat{z}_{\text{base}} > h_{\text{min}}$). However, if such a near surface cloud was not detected by the ceilometer, a mismatch would result leading to a higher RMSE.

Additionally, we exploit the stereo-derived cloud mask as a proxy of cloud cover fraction to investigate the sensitivity of the MIBase performance to the number of valid $z$ retrievals $N_{\text{val}}$, the number of $z$ retrievals marked high confidence surface $N_{\text{HCS}}$, and the number of $z$ retrievals marked high confidence cloud $N_{\text{HCC}}$ within the MIBase cell. We determine RMSE, bias, and the correlation coefficient as a function of $N_{\text{val}}$, $N_{\text{HCS}}$ and $N_{\text{HCC}}$ normalized by the total number of pixels $N_{\text{tot}}$ which the MIBase cell encloses (Fig. 8). For example, for $R_{\text{c}} = 10$ km, a total of $N_{\text{tot}} = 265$ pixel is processed by MIBase to obtain a unique $z_{\text{base}}$ retrieval. For the continental USA, most cases comprise a high portion of valid $z$ retrievals within the MIBase cell. The RMSE, bias, and the correlation coefficient are robust under different choices of $N_{\text{val}}$ and $N_{\text{HCS}}$. This suggests that MIBase generally does not depend much on cloud cover fraction. However, for cases which suggest almost apparent clear sky, indicated by high $N_{\text{HCS}}$, RMSE increases and $r$ decreases. This could be due to a lower chance of observing the same cloud in case of less extended clouds. This bias appears to strongly depend on the portion of $z$ retrievals marked high confidence cloud (Fig. 8). The increased bias for higher $N_{\text{HCC}}$ could be explained by the decreasing portion of the thin edge of the cloud compared to the thicker part of the cloud with greater horizontal extent. For instance, the edge of a larger cloud might only be partly within the MIBase cell, whereas the edge of a smaller cloud might be fully processed by MIBase. The clear increase of the bias with increasing $N_{\text{HCC}}$ shows potential for a bias correction in the future after a better understanding of the underlying reasons. The bias obtained in this study can have different sources: the different sample volumes of the defined MIBase cell and the ceilometer, biased MISR $z$ retrievals, various scene characteristics.

## 5 MIBase Application

### 5.1 Global cloud height distribution

MIBase has been applied for a three year period between 2007 and 2009 to determine the $z_{\text{base}}$ from MISR globally. Herein, $z$ data from each individual orbit have been sorted into a $0.25° \times 0.25°$ longitude by latitude grid. For each orbit and each grid box $z_{\text{base}}$ has been retrieved as described above and the median over the three year period has been calculated. Only cloud

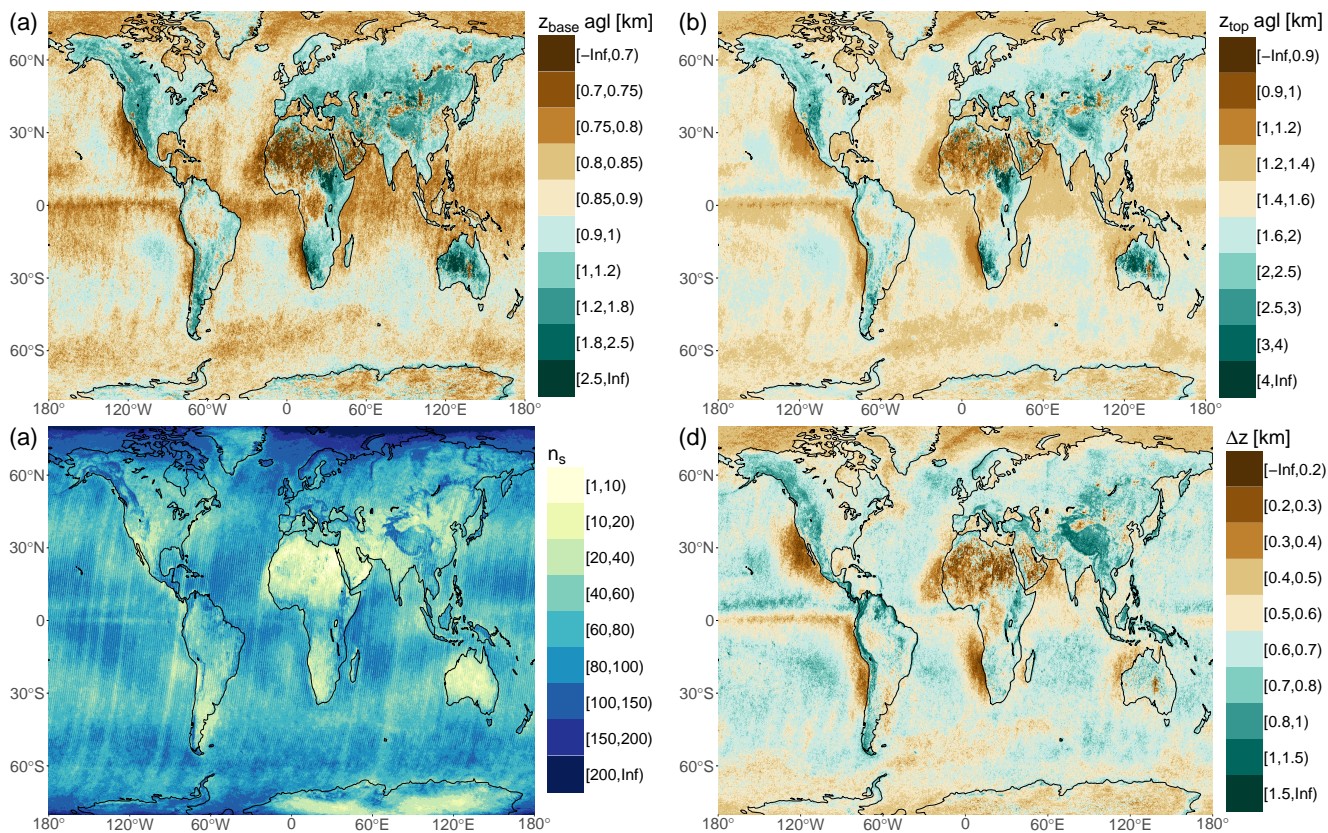

**Figure 9.** Global distribution of median cloud heights for a 3-year period (2007–2009). Shown are $z_{base}$ (a), $z_{top}$ (b), and cloud vertical extent (d) on a $0.25° \times 0.25°$ latitude–longitude grid. $z_{base}$ and $z_{top}$ are above ground level (agl). $z_{base}$ and $z_{top}$ retrievals are only included in the statistic if $z_{base}$ is below 5000 m. The number of retrievals $n_s$ (c) represents the number of valid $z_{base}$ retrievals within this 3-year period.

height retrievals below 5000 m are considered to exclude cirrus clouds from the statistics. $z_{top}$ is retrieved analogously to $z_{base}$ by applying the 95th percentile on the $z$ distribution. Taking the difference between $z_{top}$ and $z_{base}$ for each observed cloud scene yields $\Delta z$. The medians of these measures are shown in Fig. 9.

A sharp and steep gradient of the $z_{base}$ can be seen at most coast lines with a higher $z_{base}$ over land. This seems plausible as

5     boundary layers above oceans are known to be shallower. Exceptions to this rule are the Congo Basin and the Amazon Basin. These regions are moisture sinks characterized by high precipitation and excessive surface run-off. The maritime stratus cloud regions are clearly visible at the subtropical eastern boundaries of the Pacific, Atlantic and Indian ocean. These regions are characterized by prevailing high pressure due to the location at the subsiding branch of the Hadley circulation and cold ocean currents creating a temperature inversion on top of the boundary layer. For these regions cloud formation is limited to the well

10     mixed maritime boundary layer. The Intertropical Convergence Zone (ITCZ) is clearly visible in particular for the tropical Pacific ocean with a higher $z_{base}$ and even higher $z_{top}$ yielding an overall higher $\Delta z$ slightly north of the equator. Over land, this

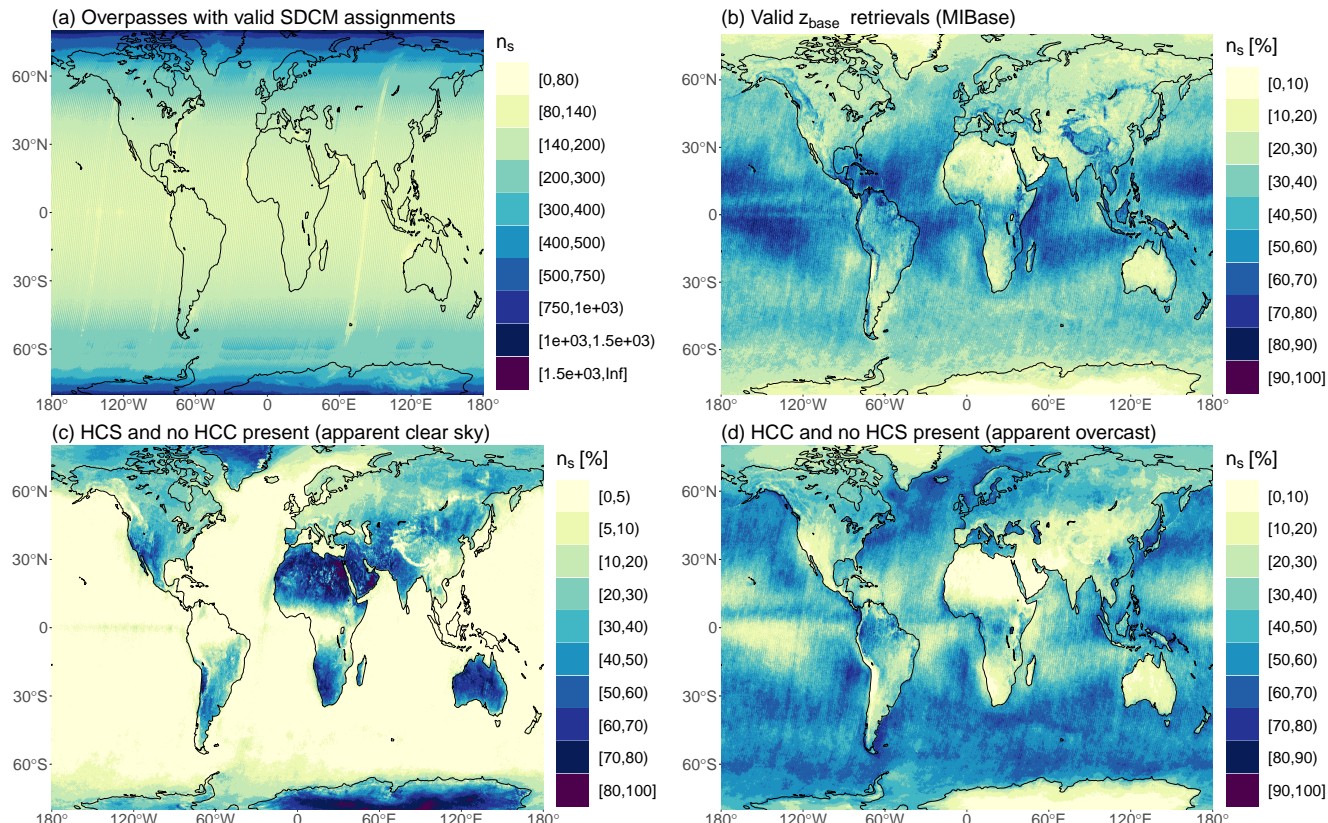

**Figure 10.** Relative occurrences of different stereo-derived cloud mask (SDCM) configurations within the three-year period (2007–2009). The reference sample size $n_s$ given in (a) corresponds to 100 % and includes all overpasses per grid cell which contain valid $z$ retrievals. (b) Relative number for which MIBase successfully retrieved $z_{base}$. (c) through (d) show the relative number of occurrence of cloud scenes which include $z$ retrievals of specific SDCM labels within a grid cell. These configurations are: (c) No high confidence cloud (HCS). These cases are apparent clear sky cases. (d) No high confidence cloud (HCS). These cases are apparent overcast cases.

phenomenon is not as clear. There, the diurnal cycle of surface heating becomes important. MISR on the Terra satellite has a morning overpass over the equator when cloud formation just begins. Taylor et al. (2017) show the diurnal cycle of cloud top temperature (CTT) derived from SEVIRI measurements indicating that the lowest $z_{top}$ occurs between 9:00 and 13:00 local time with the lowest mean CTT at 11:00. and the lowest median CTT at 12:00, close to the overpass time of MISR.

5    The sampling size varies spatially with a higher number of retrievals in the Arctic region. (Fig. 9 (c)). This is expected for a polar orbiting satellite with more frequent MISR overpasses in polar regions (Fig. 10 (a)). Generally, the causes for retrieval failure are apparent clear sky and apparent overcast situations as discussed in Section 3.4. The frequency of occurrence of such situations varies spatially. For continental dry regions in the subtropics and continental polar regions apparent clear sky conditions predominantly limit the number of $z_{base}$ retrievals (Fig. 10 (c)). The continental polar regions yield a high number

10   of cases for which the grid cell comprises only high confidence surface retrievals ($N_{HCS} = N_{tot}$, Fig. S3). This poses an even

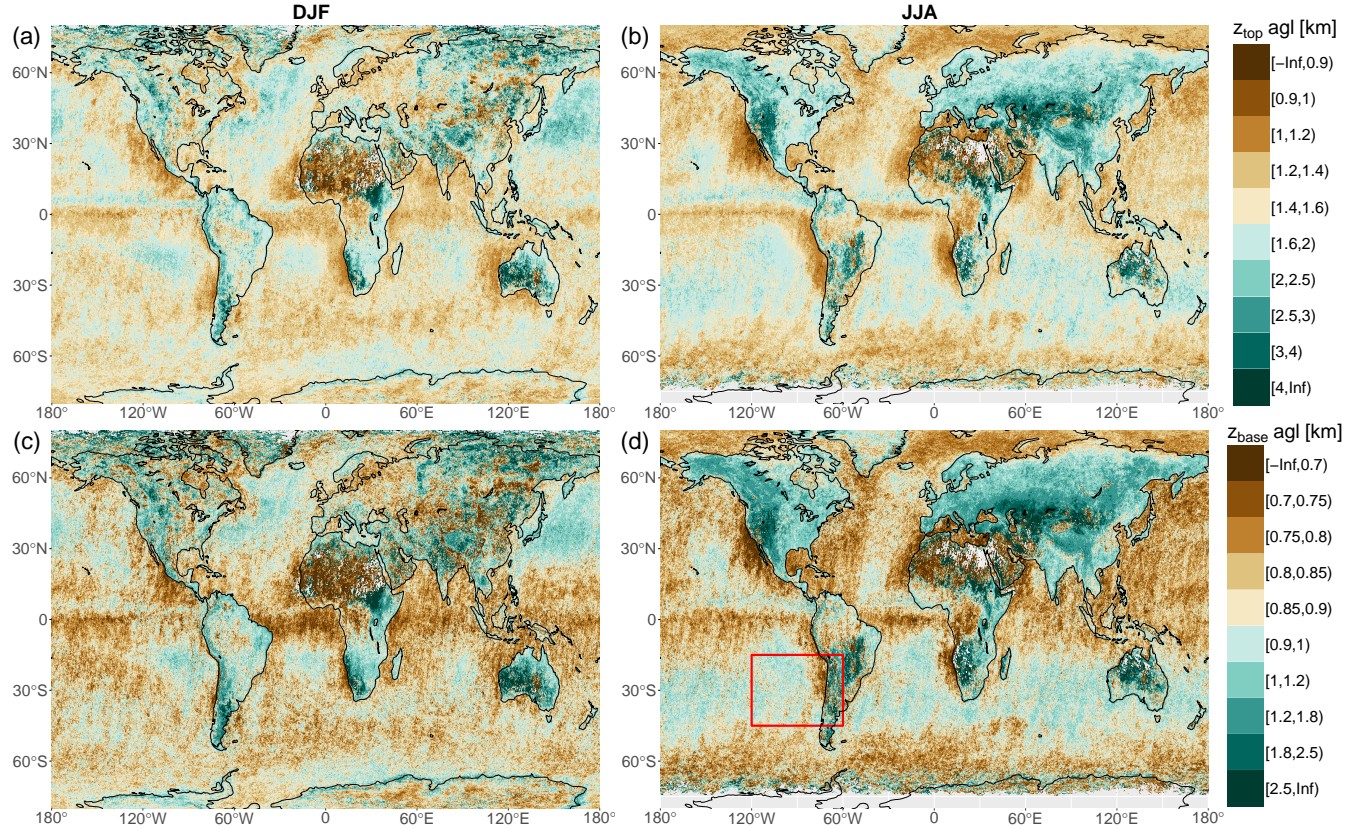

**Figure 11.** Global distribution of seasonal median cloud heights for a 3-year period (2007–2009). Shown are $z_{top}$ (a, b), and $z_{base}$ (c, d) for December, January, February (a, c) and June, July, August (b, d) on a $0.25° \times 0.25°$ latitude–longitude grid. $z_{base}$ and $z_{top}$ are above ground level (agl). $z_{base}$ and $z_{top}$ retrievals are only included in the statistic if $z_{base}$ is below 5000 m. The red rectangle in (d) frames the region for which results over a 16-year period are presented in Fig. 12.

more robust indication of apparent clear sky conditions. However, the boundary layer is typically shallower in polar regions. Therefore, boundary layer clouds occur likely below $h_{min}$, so that $z_{base}$ cannot be retrieved by the MIBase algorithm. Predominant apparent overcast conditions limit the number of $z_{base}$ retrievals for midlatitude regions over ocean and stratocumulus regions on the western boundaries of continents in the subtropics. In midlatitude continental regions, a mix of apparent clear sky and apparent overcast conditions limits the number of $z_{base}$ retrievals. In the trade cumulus regions within 30°N and 30°S, very high success rates occur (Fig. 10 (b)). A visual comparison to the 2011 mean cloud cover fraction derived from MODIS (Suen et al., 2014) indicates the plausibility of the attribution of apparent clear sky and apparent overcast.

To further investigate the plausibility of the seasonal variability of cloud heights, composites over the three year period are presented in Fig. 11. We distinguish boreal winter season comprising December, January and February (DJF) and boreal summer season comprising June, July and August (JJA). Over land and between 30°N and 70°N, $z_{base}$ and $z_{top}$ are lower during winter, when stratiform clouds prevail. In contrast, $z_{base}$ and $z_{top}$ are higher during summer, when more convective clouds are

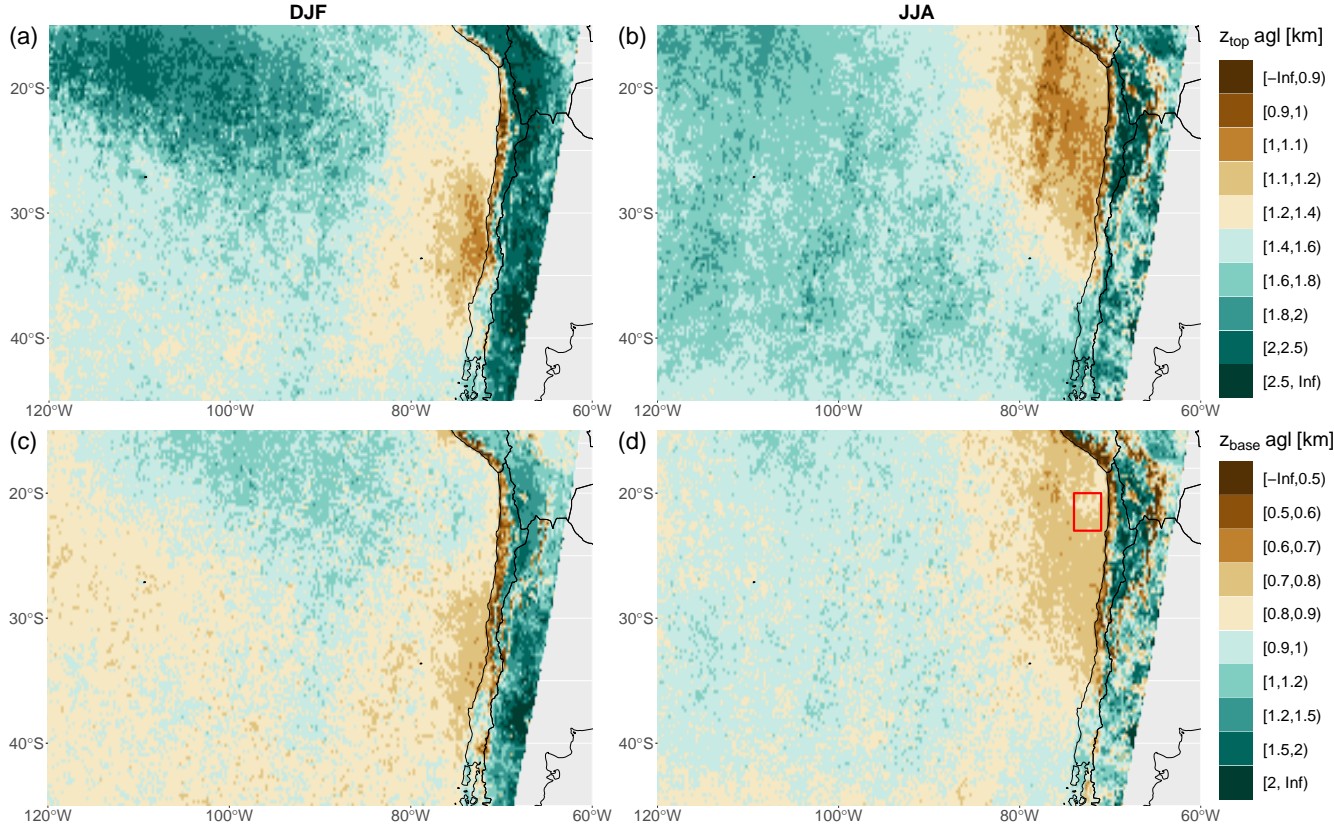

**Figure 12.** Median of $z_{top}$ (a, b), and $z_{base}$ (c, d) over a 16-year period (2001–2016) for austral summer (DJF, (a) and (c)) and austral winter (JJA, (b) and (d)) on a $0.25° \times 0.25°$ longitude by latitude grid at the southeast Pacific. $z_{base}$ and $z_{top}$ are given above ground level (agl). The red rectangle (d) frames the region for which a time series of cloud heights is presented in Fig. 13.

typically present. Boundary layer clouds are also lower during winter season since the boundary layer is shallower during the cold season. Over ocean an inverse pattern can be observed on both hemispheres. During winter $z_{base}$ and $z_{top}$ are higher than during the summer. Sea surface temperatures show less seasonal variation than air temperatures due to the higher heat capacity of the water. This causes additional instability during winter enhancing convective cloud formation which can result in higher cloud heights. Additionally, the instability during winter can be attributed to storm tracks. During summer, the influence of high pressure systems can limit convection to the maritime boundary layer causing cloud heights to be lower.

## 5.2 Southeast Pacific

The southeast Pacific hosts one of the largest and most persistent stratocumulus cloud decks on Earth as shown by Wood (2012) using data from the combined land-ocean cloud atlas database (Hahn and Warren, 2007). In this region, cloud cover and cloud

thickness have major impacts on the net cloud radiative effect, which raises the importance of studying the heights of these clouds.

Orographically induced fog at the coastal cliff ranging from Peru to northern Chile is the major source of moisture for this region (Pinto et al., 2006). $z_{\text{base}}$ and $z_{\text{top}}$ of the stratocumulus clouds near the coast determine the areas where fog can provide water to the environment at the coastal cliff. The cloud heights also affect the ability of the fog to be advected further inland across the cliff. Here, we apply the $z_{\text{base}}$ retrieval algorithm to determine the spatial and seasonal variability of $z_{\text{base}}$ and $z_{\text{top}}$ for the region (see red rectangle in Fig. 9 (bottom right)). We extend the time window to the full 16-year record of available MISR data (2001–2016). Furthermore, we investigate how well the temporal changes are represented in the global reanalysis ERA-Interim.

### 5.2.1  Spatial and seasonal variability of $z_{\text{base}}$ and $z_{\text{top}}$

For the 16-year period, the medians of $z_{\text{base}}$ and $z_{\text{top}}$ over the southeast Pacific are shown in Fig. 12. Distinguished are summer and winter season. Over ocean the median $z_{\text{base}}$ ranges from $600\,\text{m}$ near the the coast to about $1200\,\text{m}$ further west. During austral summer (DJF) the lowest $z_{\text{base}}$ is observed near the coast between 30°S and 35°S. During austral winter the region of low $z_{\text{base}}$ shifts to the north between 20°S and 30°S. This shift is in phase with the direction of the seasonal shift of the Hadley cell. It appears that the region of lowest $z_{\text{base}}$ corresponds to the strongest subsidence. During austral summer the highest $z_{\text{base}}$ clearly appear in the north, whereas during austral winter a north–south gradient is hardly visible between 120°W and 80°W. Over land, $z_{\text{base}}$ is generally higher except for the coastal line north of 35°S, where cloud heights are even lower than over ocean. There, the prevailing maritime stratocumulus clouds form orographic fog as they reach the coastal cliff. Similar spatial and seasonal patterns are apparent for $z_{\text{top}}$. Over ocean, the highest $z_{\text{top}}$ is about $2500\,\text{m}$, which is observed during austral summer in the northwest of the region. The lowest $z_{\text{top}}$ is about $1000\,\text{m}$, which is observed during winter and closer to the coast of northern Chile.

### 5.2.2  Cloud height comparison between MISR and ERA-Interim

In order to preliminarily assess how well clouds are represented in common reanalysis, we compare MISR derived $z_{\text{base}}$ and $z_{\text{top}}$ to cloud heights derived from ERA-Interim (Dee et al., 2011) which is provided by the European Centre for Medium-Range Weather Forecasts (ECMWF). Cloud heights are not a direct output variable of ERA-Interim. Therefore, the cloud liquid water content is used to infer the cloud base height $\tilde{z}_{\text{base}}$ and cloud top height $\tilde{z}_{\text{top}}$. For each grid point, the vertical column is scanned for model levels with a specific cloud liquid water content greater than $10^{-18}\,\text{kg}\,\text{kg}^{-1}$ ($\approx 0$). The bottom height of the lowest of such levels is taken as $\tilde{z}_{\text{base}}$. Moving higher in the column, $\tilde{z}_{\text{top}}$ is given by the bottom height of the next higher model level which has a cloud liquid water content equal to zero. We use data with a $0.75° \times 0.75°$ resolution, which is similar to the native grid of ERA-Interim, over a region between 20°S and 23°S and 74°W and 71°W as indicated by the red rectangle in Fig. 12. ERA-Interim data is provided 6-hourly. The comparison is performed using the 18:00 UTC output which corresponds to 14:00 Chile Standard Time (CLT). Note, MISR overpass times range around 10:51 CLT to 11:29 CLT for this particular region.

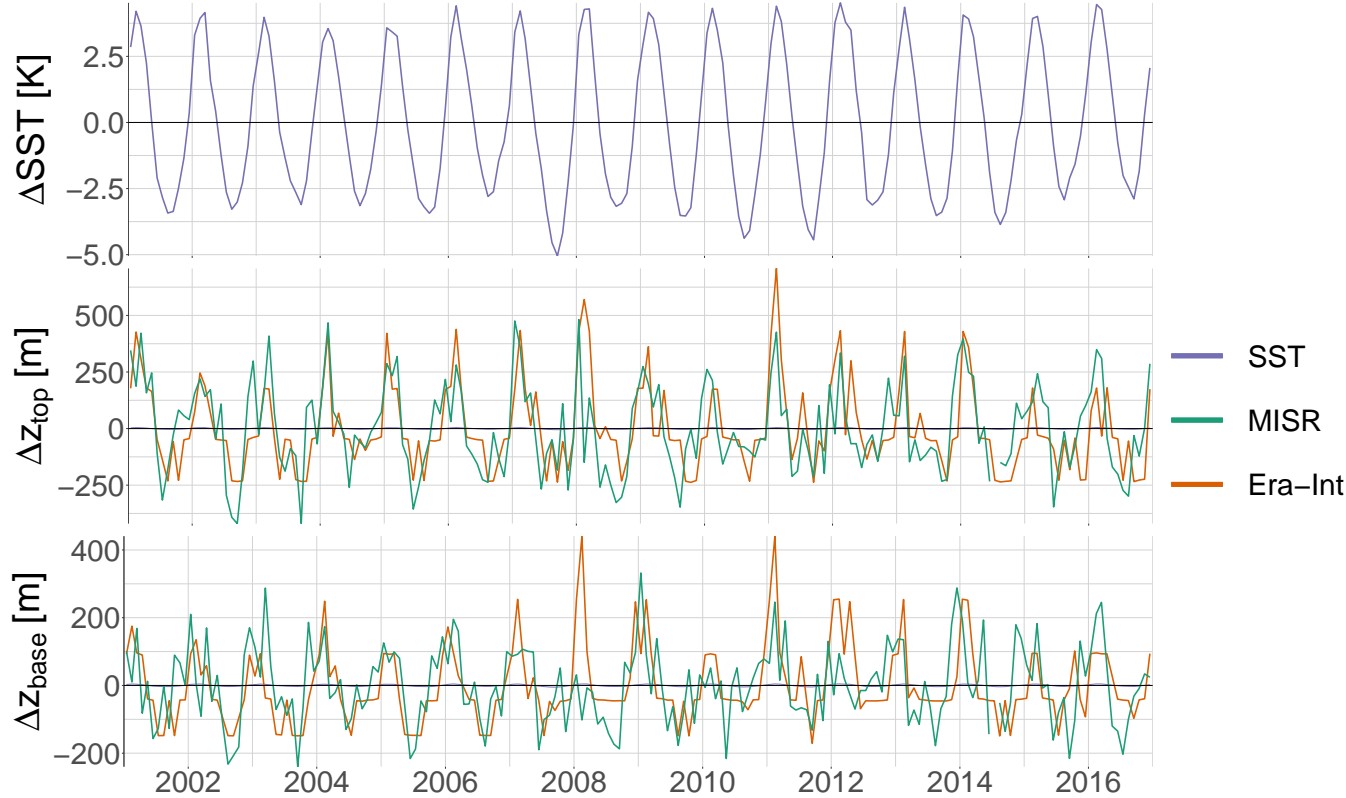

**Figure 13.** Time series of deviations of sea surface temperature $\Delta$SST (top), cloud top height $\Delta z_{top}$ (middle), cloud base height $\Delta z_{base}$ (bottom) from the corresponding mean over the entire period from 2001 through 2016. Cloud heights are derived from MISR (green) and ERA-Interim (orange). SST is derived from ERA-Interim.

For each MISR overpass and ERA-Interim 18:00 UTC output, the median cloud heights are used to calculate the median cloud heights of each month over the whole 16-year period. The mean difference of the monthly cloud heights is roughly 500 m for both cloud base height and cloud top height, with ERA-Interim yielding lower cloud heights than MISR. That $\tilde{z}_{base}$ is lower than $z_{base}$ could be due to the threshold height used to determine the MISR stereo derived cloud mask (Equation 1)

which leads to a cut-off of $z_{base}$ retrievals at $h_{min}$. At the same time the same bias is found between $z_{top}$ and $\tilde{z}_{top}$. This could be an indicator that clouds are systematically placed too low by ERA-Interim. Hannay et al. (2009) mentioned several studies which conclude that models typically underestimate the height of the planetary boundary layer (PBL) in the southeast Pacific area. This would cause boundary layer clouds to appear lower than observed. Their study compares the PBL height retrieved from in-situ measurements and remote sensing to different models. While the observations show a PBL height of 1100 m, the

models produce a PBL height between 400 m and 800 m, hence an underestimation of 700 m to 300 m. This is in accordance with the bias found here.

To reveal the annual cycle of the cloud heights, we look at anomalies from the 16-year mean of each time series (Fig. 13). These anomalies of $z_{base}$ and $\tilde{z}_{base}$ as well as $z_{top}$ and $\tilde{z}_{top}$ from their respective mean values agree rather well, thus the amplitude of the annual cycle appears very similar. Figure 13 also shows the anomaly of the sea surface temperature (SST) from its 16-year mean value. SSTs are taken from ERA-Interim as well. The peaks of the cloud heights correspond to the maxima of the SSTs. While the highest SSTs coincide with the highest cloud heights during austral summer, the lowest SSTs coincide with the lowest cloud heights during austral winter.

## 6  Conclusions

Here, we present a new method to determine $z_{base}$ over a spatial region from satellite based measurements. The MIBase algorithm derives $z_{base}$ from the high spatial resolution MISR cloud top height product $z$ if some preconditions, such as a broken cloud scene, are met. Validation against 1510 ceilometer stations in the continental USA results in a correlation coefficient of 0.66 and a RMSE of 385 m for the validation data set (year 2007). The bias of $-59$ m even states that MISR sees a slightly lower $z_{base}$ on average. This is possibly due to the larger retrieval cell which is set up for the retrievals from MISR as opposed to the point measurements provided by the ceilometer.

Very few attempts to derive $z_{base}$ from satellite have been performed and evaluated before. Desmons et al. (2013) retrieve $\Delta z$ from POLDER measurements. The standard deviation of the difference between their $\Delta z$ retrieval and reference data from CPR and CALIOP is about 964 m. However, their method is hard to compare to the MIBase algorithm, since they retrieve $\Delta z$ and make a distinction of different types of clouds which is not done in this study. The CBASE algorithm (Mülmenstädt et al., 2018) derives $z_{base}$ from CALIOP measurements even for optically thick clouds. Depending on the circumstances different retrieval uncertainties can be derived. Similar to the study presented here, they compare their $z_{base}$ retrievals with ceilometer data over the continental USA. They obtain RMSEs between 404 m and 720 m depending on the concurrent local conditions of the individual retrievals. The RMSE we obtain for the MIBase algorithm is slightly lower. Even though the two studies make use of a similar reference data base, they measure cloud heights at different times of the day. While CALIOP has an afternoon overpass, MISR has a morning overpass, when more clouds of lesser extent are present. For a more in-depth comparison and validation of the presented algorithm, more cloud height reference observations would be desirable including observations in different climate zones and especially over ocean.

Within Europe, the European Cooperation in Science and Technology (COST) activity is expected to harmonise the networks of the different weather services (e.g., Haeffelin et al., 2016; Illingworth et al., 2018), enabling more intercomparisons in the future.

An important strength of MIBase is the geometric approach which is applied to create the $z$ product from MISR measurements. Neither a calibration nor auxiliary data are necessary to obtain the $z$ product which is the starting point for the $z_{base}$ retrieval algorithm presented here. In consequence, retrievals are possible over all kinds of terrain even above ice. A disadvantage is the threshold height which MISR requires to create the stereo derived cloud mask. Therefore, depending on the terrain variability in the vicinity of the measurement, this new $z_{base}$ retrieval method is not capable of deriving $z_{base}$ below at least

560 m (flat terrain). The algorithm requires a broken cloud scene. For complete overcast within the chosen MIBase cell, $z_{\text{base}}$ cannot be retrieved. Therefore, climatologies derived from this algorithm would be biased towards cloud types for which MISR is able to observe the surface through cloud gaps.

Depending on the application, the MIBase uncertainty and the missing coverage of the diurnal cycle can be a limitation. 5 However, in combination with ceilometer networks, both temporal and spatial patterns can be investigated. The application of MIBase over a three-year period reveals plausible patterns in the global distribution and seasonal variability of $z_{\text{base}}$. A first analysis over the 16-year MISR time series in the southeast Pacific shows the potential to investigate the interannual variability of $z_{\text{base}}$. This makes MIBase a promising tool for the evaluation of climate models on seasonal and interannual time scales in data sparse regions if for example the climate model output is limited to clouds below 5 km and cloud fractions below 1 and if 10 a sufficient amount of MIBase retrievals is provided within the considered region and time period.

*Data availability.* Multiple archives providing METAR data are available. The data utilized here were downloaded from the Weather Underground archive (https://www.wunderground.com/history/airport/). The MISR Level 2TC Cloud Product data were downloaded from the NASA Langley Research Center Atmospheric Science Data Center (ftp://l5ftl01.larc.nasa.gov/MISR/MIL2TCSP.001/). ERA-Interim data were downloaded from the ECMWF data server via Web-API. The MIBase cloud base dataset (Böhm, 2019) is freely available at the Collaborative Research Centre 1211 database under the DOI https://doi.org/10.5880/CRC1211DB.19. It comprises $z_{\text{base}}$ retrievals globally on a $0.25° \times 0.25°$ grid for a three year period (2007–2009). Daily files include $z_{\text{base}}$ retrievals derived from the MISR MIL2TCSP product for about 14 respective Terra revolutions around the Earth. Cloud base altitudes are given above the WGS 1984 ellipsoid. Furthermore, the surface altitude is provided to derive the cloud base height above ground level.

## Appendix A: Sensitivity to threshold height $h_{\text{min}}$

The distinction between surface and cloud retrieval according to the threshold height described by Equation 1 introduces a constraint to the $z_{\text{base}}$ retrieval algorithm. Below a height of 560 m for flat terrain, or higher for more complex terrain, $z_{\text{base}}$ retrievals are not possible. As an attempt to lower this threshold height, we adjusted $H_{\text{SDCM}}$ in Equation 1, so that:

$$h_{\text{min}} = 300\,\text{m} + H + 2\sigma_{\text{h}} \tag{A1}$$

This modification results in a bimodal retrieval density clearly showing a mode consisting of surface retrievals (Fig. A1). 25 Therefore, the original threshold height given by MISR has to be applied, in order to ensure that only cloud retrievals are utilized during data processing.

*Competing interests.* The authors declare that they have no conflict of interest.

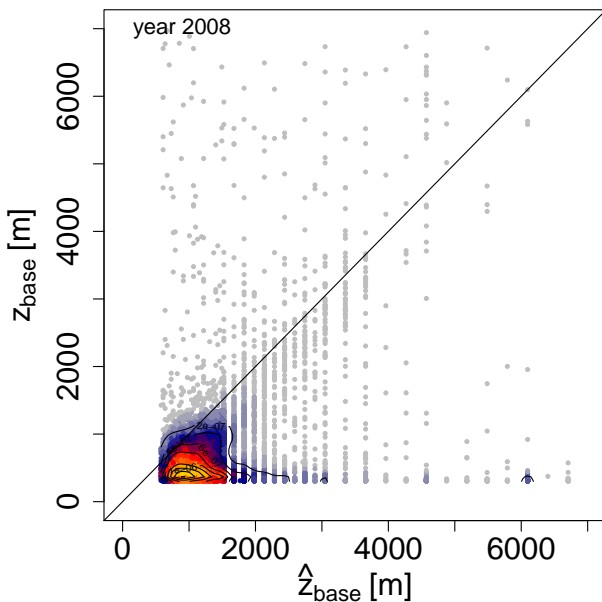

**Figure A1.** Joint density of $z_\text{base}$ and $\hat{z}_\text{base}$ for the year 2008 applying a lower threshold height $h_\text{min} = 300\,\text{m} + H + 2\sigma_\text{h}$ (Equation A1) for the distinction between surface and cloud pixels in contrast to Equation 1.

*Acknowledgements.* The MISR Level 2TC Cloud Product data were obtained from the NASA Langley Research Center Atmospheric Science Data Center (Mueller et al., 2013). We gratefully acknowledge financial support by the Deutsche Forschungsgemeinschaft (DFG, German Research Foundation) – Projektnummer 268236062 – SFB 1211. We thank the six reviewers for their constructive feedback.

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
