# Peer review of "Cloud base height retrieval from multi-angle satellite data"

_Atmospheric Measurement Techniques, 2018_

## Referee Comment (RC1) · Anonymous Referee #3 · 9 Oct 2018

The paper presents an interesting technique to infer cloud base height from the MISR standard cloud product. It demonstrated a valuable skill with this technique that uses the 15 percentile threshold to the vertical distribution of MISR cloud heights in a 10-km domain. The algorithm can be readily applied to all MISR cloud data for seasonal and global statistics of cloud base height.

The technique is perhaps valid for broken cloud scenes in the 10km domain, but would fail if clouds are 100$ optically thick or overcast in the domain. This is often the case over land, but not necessary over ocean. The authors should acknowledge this limitation in the abstract and conclusion.

For the sensitivity calculations summarized in Table 3, the results might be dependent on roughness of terrain since MISR cloud height retrievals would correlate worse with

ceilometer base height if the site is surrounded by mountains. What would be the results if only those sites with flat terrain (in 10, 20, or 30 km radius) are included in statistics?

Some minor issues and English: p4, line 12: MISR cloud motion vector in L2TCSP file is determined at 17.6 km resolution, and is used to derive H_SDCM by correcting the wind-induced parallax effect. As noted in Mueller et al. (2013, 2016), the cloud height and along-track wind errors are correlated.

p17, line 21 .. shows a higher number of ...

p.17, line 27 .. seasons ..

p20, line 20 ... mentioned ...

---

## Author Comment (AC1) · 19 Oct 2018

"The paper presents an interesting technique to infer cloud base height from the MISR standard cloud product. It demonstrated a valuable skill with this technique that uses the 15 percentile threshold to the vertical distribution of MISR cloud heights in a 10-km domain. The algorithm can be readily applied to all MISR cloud data for seasonal and global statistics of cloud base height."

We thank the reviewer for her or his constructive comments and suggestions.

"The technique is perhaps valid for broken cloud scenes in the 10km domain, but would fail if clouds are 100 optically thick or overcast in the domain. This is often the case over land, but not necessary over ocean. The authors should acknowledge this limitation in

the abstract and conclusion."

Yes, in case of complete overcast, MIBase cannot derive the cloud base. The algorithm requires at least one MISR z pixel within the field of view which is a surface pixel according to the stereo derived cloud mask. This is stated in the abstract: "[...] It can be applied if some cloud gaps occur within the chosen distance [...]", but not again in the conclusion. We can fix this. Cloud optical thickness itself should not be a limitation. However, the cloud shape could be a limitation. The thinner cloud edge should be visible by the instrument for the algorithm to retrieve the base height (see page 7, line 4.).

"For the sensitivity calculations summarized in Table 3, the results might be dependent on roughness of terrain since MISR cloud height retrievals would correlate worse with ceilometer base height if the site is surrounded by mountains. What would be the results if only those sites with flat terrain (in 10, 20, or 30 km radius) are included in statistics?"

Interesting point. To derive a quantity to judge what is "flat" terrain, we use the standard deviation of the average scene elevation (ASE):

- ASE from the MISR Ancillary Geographic Product (Bull et al, 2011) at 1.1 km resolution (like MISR cloud top height product)
- use ASE within field of view around each ceilometer station ( $R_{fv} = 5 \text{ km}$ , 10 km, 15 km, 20 km, 30 km)
- calculate standard deviation of ASE for each ceilometer station and each  $R_{\rm fv}$
- plot density of standard deviations for each  $R_{\rm fv}$
- filter out stations with higher standard deviation to calculate statistics (correlation r, bias, RMSE)

Results are shown in Fig. 1. On the bottom row of the plot you see "max allowed sd" on the abscissa. This means, ceilometer stations with higher standard deviation (sd) then the given value have been excluded from the statistics. "Inf" means all stations are considered, as they have been before.

The threshold values of 20 m, 30 m, 40 m, 50 m, and 60 m for sd correspond to the 74th, 84th, 89th, 92th, and 94th percentile of the distribution of sd, respectively, for a radius of 5 km. For a radius of 30 km, the corresponding percentiles are 30th, 49th, 60th, 66th, and 71th. In other words, for a larger radius, more stations are excluded for the same sd threshold.

The correlation increases slightly if the more varying terrain is excluded. But only until sd=50m. If an even stricter threshold is applied the correlation decreases again. The bias improves more or less monotonically with a stricter sd threshold, i.e. for "flatter" terrain. The RMSE improves as well.

Conclusion: Limiting the comparison to sites showing more homogeneous terrain improves the comparison slightly.

"Some minor issues and English: p4, line 12: MISR cloud motion vector in L2TCSP file is determined at 17.6 km resolution, and is used to derive H\_SDCM by correcting the wind-induced parallax effect. As noted in Mueller et al. (2013, 2016), the cloud height and along-track wind errors are correlated. p17, line 21 .. shows a higher number of ... p.17, line 27 .. seasons .. p20, line 20 ... mentioned ..."

Thank you for these points. We will take them into account and adjust the manuscript accordingly.
Interactive

comment

20 30 40 50 60 Inf max allowed sd [m]

**Fig. 1.** (a) Density of the occurring std. deviations for various radii. Recalculation of corr. coeff. r (b), bias (c), RMSE (d) using only ceilometer stations below a threshold sd as denoted on the abscissas.

---

## Referee Comment (RC2) · Anonymous Referee #4 · 22 Oct 2018

This paper describes an interesting technique to infer cloud base height from MISR measurements within selected areas. The paper is well written. The technique is described well. I recommend the paper to be accepted for publication after some minor suggested additions and corrections listed below.

General comment:

The authors should describe better to which kind of cloud fields this method can be applied. The abstract states "it can be applied if some cloud gaps occur within the chosen distance of typically 10 km." However, cirrus are excluded in the evaluation, because it probably would not work on cirrus. I also do not expect the technique to work particularly well on areas dominated by deep convection and congestus, for example. Please discuss the expected limitations of the technique related to cloud types.

Specific comments:

Page 7, line 7: I agree that MISR cloud top heights are probably superior to those of other passive satellite instruments, but not to those from active instruments, in particular lidar.

Figure 12: In the caption note that these are anomalies. Also add a Delta in front of the y-axis labels.

I thought the discussion of multi-layer situations on page 8 was interesting and I suggest to add some words about that in the conclusions.

---

## Referee Comment (RC3) · Anonymous Referee #5 · 30 Oct 2018

The attempt to derive cloud base heights from MISR data is interesting, but as far as I can tell they basically take the minimum retrieved cloud height, assume it corresponds to the base height and move on from there. The authors need to state more clearly that this algorithm is only valid over broken clouds, and indeed I would be very interested in seeing a study of the accuracy of the results as a function of scene structure and degree of brokenness, and also as a function of the number of unobscured cloud top and side pixels as available in the MISR TC\_ALBEDO product. I am willing to reconsider the paper if the authors perform such a study as I think that would be much more interesting than just the minor algorithm parameters such as R, N and P.

Additionally they need to clarify which MISR product they are using (TC\_STEREO or TC\_CLOUD), and which type of SDCM (WindCorrected or WithoutWindCorrection). I

am unsure if they are using Stereo or Cloud, because they mention the correct short name for Cloud, but list the wind resolution as being 70.6 km and Stereo is at 70.4 km and Cloud retrieves its winds at 17.6 km. I am hoping this is just a typo on their part but I'm not sure. It is my opinion that they need to use the WindCorrected heights from the TC\_CLOUD product.

---

## Referee Comment (RC4) · Anonymous Referee #6 · 2 Nov 2018

The authors propose a method to derive cloud base height from MISR measurements. Here, they make use of the 9-angle viewing capabilities of the instrument and derive all possible cloud top heights within a specified area, the (approximately) lowest $z\_top$ is then attributed to be the base height of the cloud field within the specified area. For this algorithm to work, several preconditions have to be met, as specified by the authors. First, the cloud field has to be inhomogeneous so that MISR can see thin cloudy layers around the cloud field's edges. Second, it should not be used for thin cirrus. Personally I would say it will probably also have problems in regions with very inhomogeneous cloud bases or in regions with strong convective systems which means very inhomogeneous but also very thick clouds. Due to these restrictions I am not convinced that this product will be an easy-to-use tool for the quantitative assessment

of cloud base height in climate models as stated in the conclusions. However, the comparison to METAR data shows good results. The article is well written and the method is clearly explained.

Nevertheless, I think it could be improved because a better analysis of the situations in which the retrieval does not perform well would be necessary in order to evaluate its capabilities.

Also some statistics that quantify, in how many cases the algorithm could not retrieve a cloud base height is missing. These values should be given for each possible retrieval rejection, a too homogeneous cloud cover for instance, in comparison to the number that would have theoretically been possible.

In Fig. 9 b), the ITCZ should be more visible in the Atlantic Ocean and over Africa, there are almost no $z\_top$ values over 1.4 km. Even if the analysis is restricted to cases with $z\_top < 5000$ m, I would assume that there should be more $z\_tops$ higher than 1.4 km. Could you please comment on that? And why is $z\_top$ restricted to 5000 m, is this threshold not only applied to $z\_base$ in order to exclude cirrus?

Fig. 9 a): Since the number of valid retrievals over the Sahara is so small, it is quite understandable, that the cloud base height jumps between very small and very high values and a warning is given by the authors on page 17. In order to use maps of this kind for a climate model evaluation, many more valid data points would be necessary. This should be noted in the conclusions.

Fig. 9c): Why is the sample size low over Antarctica? Shouldn't it be covered with approx. 50% cloud cover throughout the year?

Some minor comments:

p 4, l 19: please specify "SDCM" in H_SDCM

p 16, l 6: "yielding an overall higher" something is missing here.

P 17, l 1: Do you refer to Fig. 9 c) instead of b)?

---

## Referee Comment (RC5) · Anonymous Referee #2 · 5 Nov 2018

The authors describe a novel algorithm for the retrieval of cloud base height (CBH) from MISR satellite measurements. Global information on cloud base height is important for many applications and the retrieval approach is interesting and promising. However, the manuscript is not sufficiently convincing in demonstrating the reliability of the new CBH product. Below are a number of major issues to be addressed before this manuscript may be suitable for publication.

**General comments**

The algorithm is tuned with METAR observations over the U.S., i.e. extratropical land surface. How representative is this for ocean surfaces and other climate zones, where different cloud types prevail? To show the skill in other regions, some comparisons with independent measurements elsewhere would be required.

More information on the success rate of the retrieval algorithm is required to evaluate how useful it is. Statistics of the number of samples n\_s are given but these are only absolute numbers, not (fractional) success rates. For example, Table 3 indicates that n\_s is between 3059 to 7772 depending on R\_fv. A rough calculation based on 1510 ceilometers, a MISR revisit time of 6 days, and a cloud fraction of 50% would potentially yield around 45,000 cloudy collocations. This suggests that in only 10% of the cloudy cases, a valid CBH retrieval is obtained. Is this correct? Such statistics, accompanied by the relative occurrence of different causes of retrieval failure, need to be provided, also for the global plots, to evaluate the applicability of the method.

The calibration of the algorithm is done for z\_base smaller than 3000m, because of the limited range of the ceilometers. However, for the global composites an upper threshold of 5000m is used. It is unclear whether this extrapolation outside the range for which the product has been trained, is valid.

The global maps in Figs. 9 and 10 are hard to interpret because upper limits of z\_base and z\_top have been applied. What does the median of a distribution cut-off at some value tell us? I'm also confused by the description of Fig. 9, which says that the ITCZ is clearly visible with higher z\_base and z\_top. In the plots a brown band can be seen, but these are lower rather than higher values. Can you explain? Is it also possible that the results in these multi-year median are biased to certain cloud types? For example, in the stratocumulus (Sc) areas west of the continents, cases with closed Sc will probably not yield a valid retrieval, while for open Sc z\_base can be retrieved, so that the end result will be biased to the latter.

The authors define percentile (P) values of the MISR lowest cloud layer z distribution to obtain z\_base and z\_top. For z\_base one would expect P=0 because z\_base should be lower than any MISR-derived cloud-top height. The chosen value P=15 is motivated by the noise in MISR z, which makes sense. However, for z\_top I do not understand the chosen value P=95. All MISR z values are actual estimated cloud top heights. The logical way to aggregate these is to average the individual z observations or take the

AMTD
median. In other words, a value P=50 seems natural. The choice of P=95 should thus be motivated.

Cloudsat is, especially in combination with Calipso, arguably the most accurate source of cloud base height (as well as cloud top height) information from space. Surprisingly, Cloudsat is not mentioned at all (except for two remarks in the context of the Desmons et al. paper) in the manuscript. At the least, Cloudsat should be discussed, and it would also be good to make some comparisons with this instrument, even if direct collocations with MISR supposedly only occur at high latitudes.

Specific comments

P1, Abstract: The abstract does not include any information on the cloud types the algorithm is applicable to. In the manuscript this information is also too limited. Does the method work for cirrus, or for deep convective clouds?

P2, L8: Stephens et al. (2002) is mainly about Cloudsat. It's not the appropriate reference for CALIPSO.

P4, L4: Is 'aftward' correct English?

P5, L6: 'measurements': what is measured?

P5, L9: Is the value 5000 ft correct? It seems such a big jump from 100 ft and 200 ft in the two lower height categories, respectively.

P5, L10: This suggests that bins is the same as clusters, which is not the case, I assume.

P8, L1-3: This paragraph looks out of place here. Suggest to move it somewhere else.

P8, L7: Suggest to replace 'the estimated' by 'a typical'.

P8, L15: what are 'z pixels'?

P8, L25-26: Does this mean that the case in Fig. 4 is not included in the statistics?
Isn't it a bit strange to present a case study that is not part of the selection applied furtheron?

P10, L5: Fig. 4 includes only one 'h\_gap'.

P10, L13-14: The second layer detected by MISR has a base height between 5000 and 5500 m a.s.l. The ceilometer detects layer base heights at 853 m, 2286 m, and 7010 m a.g.l. None of these seems to match with the second MISR layer. Can you explain?

P12, Fig. 6: I assume this figure is for N=10 and R\_fv=10 km. Can you add this to the caption for completeness?

P12, L6: N=10 seems a relatively low number and one could wonder whether P=15 makes sense for such a small N. Can you comment?

P13, Fig. 7: Can you add a color bar? Is the scale linear or logarithmic?

P14, Table 4: Do you have any explanation why 2007 has 30% more valid retrievals than 2008?

P14, L5: Certainly the different measurement geometry (point over time versus circular area instantaneous) can cause differences. But why would this lead to a bias, and why to a negative bias of MISR in particular? Can't you tune the overall bias to near zero by increasing P?

P15, Fig. 8: I'm not sure how useful this distinction in two geometrical thickness classes is, in particular because this thickness is based upon the MISR retrieval itself.

P15, L3: The termination of the z\_base range by the threshold height relates mostly to the lower thickness class, so it would be better to write: 'The smaller E for clouds with a smaller Delta z ..'. P16, L6: Sentence ends unexpectedly.

P17, L1: The sampling size is in Fig. 9c.
P19, L3: effect should be affect.

P19, L32: A mean difference of 500 m is quite large relative to the retrieved z\_base in Fig. 11, which appears to vary between 800 and 1200 m for the selected region. Is it reasonable to assume that the model can simulate a reasonable seasonal variation of  $z_base$  if it has such a large bias?

---

## Referee Comment (RC6) · Anonymous Referee #7 · 30 Nov 2018

As there are already five reviews available (and more referees have accepted the review of the paper) I can be short with my statements. I agree with what has been mentioned by the other reviewers with respect to the pre-conditions (cloud optical thickness, homogeneity), so I can restrict myself to comments mainly related to the ceilometers as this has not yet been covered in detail.

1. Section 2.1: The expression "$z$ pixel" might be revised/improved.

2. Section 2.2: Please add 1–2 sentences to describe the type of ceilometers used, and the basic characteristics of the instrument and the cloud height retrieval. Is the very coarse vertical resolution of the METAR-messages an issue? What about using backscatter profiles from ceilometer networks, e.g. in Europe: de-

rived cloud base heights are quite reliable and the vertical resolution is in the order of 10 meters. Please comment on this; maybe in the conclusions. Is the variability of the 30 s messages used to exclude certain data sets (temporal variability translated to spatial inhomogeneity [taking into account the bins of the messages])? The discussion of the implications of the time period of 30 minutes for averaging could be extended.

3. Section 3: Better use another word for "field of view" ($R_f$) here: according to page 7, line 11 it has nothing to do with the optics of the radiometers onboard of MISR as one might expect.

4. Section 3.2: Taking into account the very poor vertical resolution of the ceilometers and the large "footprint" of the inter-comparison I feel that it is not justified to end up with a $\hat{z}_{\text{base}} \approx 853$ m (pretending a one-meter-accuracy). Can you give an uncertainty instead of using "$\approx$".

   Page 10, line 17 states that a cloud base height of 7010 m was retrieved. In section 2.2 it is stated that the ceilometers have a vertical range of up to 3700 m. Please explain.

5. Section 5.2.2:

   The caption of Fig. 12 could be misleading. Mention that deviations are shown right at the beginning of the text.

   The conclusions of the papers cited in Hannay et al. (2009) are mainly based on thermodynamics. They do not cover pbl-retrievals based on backscatter. This is however relevant for ceilometers (that are used as reference in this paper). Therefore the agreement/disagreement of ceilometer-retrievals with model results should be discussed as well: a lot of papers have recently been published focussing on the potential of ceilometers in general and the determination (and its accuracy) of the mixing layer height (or pbl).

6. Section 6:

   I agree that the MIBase can be a promising tool for remote areas, and for climatological studies with the corresponding (extended) spatiotemporal averages. Nevertheless a few comments on the benefit of the retrieval based on individual observations would be desirable, considering the large uncertainty and the missing coverage of the diurnal cycle. So combination with ground based ceilometer networks (where available) should be envisaged, especially as ceilometers are a very direct and accurate approach (no calibration required, continuous operation) for $z_{\text{base}}$-retrievals.

---

## Author Comment (AC2) · 1 Feb 2019

*"This paper describes an interesting technique to infer cloud base height from MISR measurements within selected areas. The paper is well written. The technique is described well. I recommend the paper to be accepted for publication after some minor suggested additions and corrections listed below."*

Thank you for your constructive feedback. We have addressed your comments in the following way:

*"The authors should describe better to which kind of cloud fields this method can be applied. The abstract states "it can be applied if some cloud gaps occur within the*

*chosen distance of typically 10 km." However, cirrus are excluded in the evaluation, because it probably would not work on cirrus. I also do not expect the technique to work particularly well on areas dominated by deep convection and congestus, for example. Please discuss the expected limitations of the technique related to cloud types."*

We agree that MIBase has limitations in respect to cloud types. Therefore, we introduced two new sections "3.4 Scene limitation" and "4.1 Scene structure influence" into the manuscript and modified abstract and conclusions accordingly. In short: A bias towards a certain cloud type can be introduced by two sources: the MISR cloud top product, yielding more valid retrievals for specific cloud types, or the MIBase algorithm. Therefore, these two aspects would have to be investigated separately. Many approaches to distinguish different cloud types from satellite data have been proposed, e.g. the cloud optical depth / top height approach by the International Satellite Cloud Climatology Project (ISCCP). However, this kind of classification is not unique, depends on the horizontal resolution and likely needs additional data products. Therefore, it goes beyond the presented study.

To investigate the performance of the MIBase algorithm in dependence on parameters which are also relevant for cloud type classification, we determine RMSE, bias, and the correlation coefficient in dependence on $z_{top}$ and the cloud vertical extent $\Delta z$. A figure showing the results can be found in the added supplement material (Fig. 2). For most cloud fields which are observed within this study, $z_{top}$ ranges between $1000\,\mathrm{m}$ and $2000\,\mathrm{m}$ (supplement Fig. 2a). For a lower $z_{top}$, the RMSE shows a minimum of approximately $300\,\mathrm{m}$ and increases for clouds with higher $z_{top}$. As we already discussed at the end of section 4, this behavior could be due to the termination of the $z_{base}$ range by the threshold height $h_{\min}$. However, in case of even lower $z_{top}$ values, the RMSE increases. As these low $z_{top}$ values approach the threshold height, two different cloud scenes are possible: the cloud extents below the threshold height indicating near surface clouds or fog, or the cloud is extremely thin. In this study we excluded scenes for which the ceilometer reported a cloud below the threshold height. Therefore, low clouds or fog

should not be included in the statistics, unless the ceilometer did not detect it. In particular for very thin clouds, the RMSE is lowest (supplement Fig. 2f). In conclusion the higher RMSE for very low $z_{top}$ values could indicate that the MIBase algorithm does not perform as well in proximity to the threshold height. This is also indicated by an increasing correlation coefficient with increasing $z_{top}$ (supplement Fig. 2d). High correlation coefficients (supplement Fig. 2h) and low RMSE (supplement Fig. 2f) for cloud thicknesses up to $1000$ m indicate that the algorithm works particularly well for thinner clouds.

For cirrus clouds or high clouds in general, we cannot make a robust statement as we tax the accuracy of the METAR reports insufficient for this particular height range.

For overcast situations, $z_{\text{base}}$ cannot be retrieved. We added statistics on how many apparent overcast cases are observed to the manuscript (Tab. 5, Fig. 10 of the revised manuscript).

*"Page 7, line 7: I agree that MISR cloud top heights are probably superior to those of other passive satellite instruments, but not to those from active instruments, in particular lidar."*

Agreed. We added "passive" to the sentence.

*"Figure 12: In the caption note that these are anomalies. Also add a Delta in front of the y-axis labels."*

Thank you for pointing out that this was not clear enough. We added a Delta to the y-axis label and edited the caption to make it clearer to the reader that we show anomalies here.

*"I thought the discussion of multi-layer situations on page 8 was interesting and I suggest to add some words about that in the conclusions."*

[Figure]

If we include multi-layer cases we would add 689 cases (10%) to the statistics for the year 2007 (432 cases or 8% in 2008). With these additional cases, the correlation with the ceilometer retrievals decreases slightly from 0.66 to 0.64 (2007 and 2008) and the RMSE increases slightly from $385\,\mathrm{m}$ to $395\,\mathrm{m}$ (2007) and from $404\,\mathrm{m}$ to $418\,\mathrm{m}$ (2008). The MISR cloud top height product includes a correction for cloud advection. This is carried out via a cloud motion vector which is determined at a certain cloud feature height at a $17.6\,\mathrm{km}$ horizontal resolution. The wind correction is applied to any cloud top height retrieval which is within $840\,\mathrm{m}$ distance from the feature height of the cloud motion vector. Collocated cloud motion vectors and their neighbors are considered for the correction of a cloud top height retrieval. We suspect that the wind correction in multi-layer cases, i.e. cases with a wide range of cloud heights, might not be as accurate. At the same time multi-layer cases might also lead to a false comparison with the ceilometer, since it is unclear which layer passed over the ceilometer and which may have not. Therefore, we decided to exclude multi-layer cases from the evaluation and do not mention them in the conclusion.

Please also note the supplement to this comment:
https://www.atmos-meas-tech-discuss.net/amt-2018-317/amt-2018-317-AC2-supplement.pdf

———————————————

---

## Author Comment (AC3) · 1 Feb 2019

Thank you for your constructive feedback. We have addressed your comments in the following way:

"The attempt to derive cloud base heights from MISR data is interesting, but as far as I can tell they basically take the minimum retrieved cloud height, assume it corresponds to the base height and move on from there. The authors need to state more clearly that this algorithm is only valid over broken clouds, [...]"

As also other reviewers commented on the dependence on cloud types, we introduced two new sections "3.4 Scene limitation" and "4.1 Scene structure influence" into the

manuscript and modified abstract and conclusions accordingly. Specifically we added: "The occurrence of a broken cloud field is a basic assumption of MIBase." to Section 3.1.

"[...] indeed I would be very interested in seeing a study of the accuracy of the results as a function of scene structure and degree of brokenness, [...]"

Figure 8 in Section 4.1 of the revised manuscript shows the dependence of RMSE, bias and correlation coefficient on the configuration of the stereo-derived cloud mask. In particular, the dependence on the number of z retrievals marked high confidence cloud within the considered cloud field can serve as a proxy for cloud cover fraction.

"[...] and also as a function of the number of unobscured cloud top and side pixels as available in the MISR TC\_ALBEDO product. I am willing to reconsider the paper if the authors perform such a study as I think that would be much more interesting than just the minor algorithm parameters such as R, N and P."

The TC\_ALBEDO (MIL2TCAL) product provides the number of unobscured top and side pixels at a resolution of 2.2 km. This means, the number of pixels at the actual MISR resolution (275 m) within a 2.2 km area which observe the same reflecting layer are counted. Therefore, the product might suffice as an indicator of a more or less complex scene structure. As the influence of the scene structure on the MIBase performance has been brought up also by other reviewers, we decided to extent the discussion on this (new Section 4.1). Instead of using the MIL2TCAL product to further investigate the scene structure, we decided to exploit the stereo-derived cloud mask in more detail because it is already included in the MIL2TCSP product which builds the base for MIBase. Furthermore, we investigated the influence of  $z_{top}$  and  $\Delta z$  on the the performance of MIBase. These parameters are also characterizing the scene structure in more detail. An additional figure is included in the supplement (Fig. 2).

"Additionally they need to clarify which MISR product they are using (TC\_STEREO or TC\_CLOUD), and which type of SDCM (WindCorrected or WithoutWindCorrection). I

am unsure if they are using Stereo or Cloud, because they mention the correct short name for Cloud, but list the wind resolution as being 70.6 km and Stereo is at 70.4 km and Cloud retrieves its winds at 17.6 km. I am hoping this is just a typo on their part but I'm not sure. It is my opinion that they need to use the WindCorrected heights from the TC\_CLOUD product."

Thank you for pointing this out. Unfortunately, a typo occurred which has been corrected. As we state in the manuscript, we are using the MISR Level 2TC Cloud Product (MIL2TCSP) which provides the cloud top height and the stereo-derived cloud mask at a 1.1 km resolution with and without wind correction. Here, we are using only the wind corrected data sets. As stated in the MISR Level 2 Cloud Product Algorithm Theoretical Basis (Mueller et al., 2013) the wind correction is carried out via a Cloud Motion Vector which is determined at a resolution of 17.6 km, like you mention. We added a sentence about the wind correction to Section 2.1 to make this clear.

Please also note the supplement to this comment: https://www.atmos-meas-tech-discuss.net/amt-2018-317/amt-2018-317-AC3supplement.pdf

СЗ

---

## Author Comment (AC4) · 1 Feb 2019

"The authors propose a method to derive cloud base height from MISR measurements. Here, they make use of the 9-angle viewing capabilities of the instrument and derive all possible cloud top heights within a specified area, the (approximately) lowest z\_top is then attributed to be the base height of the cloud field within the specified area."

Thank you for your constructive feedback. We have addressed your comments in the following way:

"For this algorithm to work, several preconditions have to be met, as specified by the authors. First, the cloud field has to be inhomogeneous so that MISR can see thin

cloudy layers around the cloud field's edges. Second, it should not be used for thin cirrus. Personally I would say it will probably also have problems in regions with very inhomogeneous cloud bases or in regions with strong convective systems which means very inhomogeneous but also very thick clouds. Due to these restrictions I am not convinced that this product will be an easy-to-use tool for the quantitative assessment of cloud base height in climate models as stated in the conclusions."

We agree that MIBase has limitations in respect to cloud types. Thin cirrus will be problematic because the MISR clout top height retrieval method is based on frequencies in the visible light range, for which thin cirrus is hard to detect. Therefore, a height limit of 5 km is used for the global application. Heterogeneous cloud base heights pose a challenging scene as well, since we assume that the lower end of the cloud top height distribution is representative for the cloud base height within the region of interest. However, any kind of retrieval method may have trouble with heterogeneous cloud base heights. In the new Section 4.1 "Scene structure influence", we included an investigation of the MIBase performance in dependence on  $\Delta z$  and  $z_{\rm top}$  (supplement Fig 2). In short, MIBase performs best for shallow low clouds.

We agree that some constraints have to be taken into account when using MIBase to evaluate cloud base height in climate models. MIBase could still be a valuable tool, if for example the climate model output is limited to clouds below 5 km and cloud fractions below 1. While the comparison of individual clouds suffers from the large uncertainty, evaluation on seasonal and inter-annual scales should yield robust results. We modified the conclusion accordingly.

"However, the comparison to METAR data shows good results. The article is well written and the method is clearly explained.

Nevertheless, I think it could be improved because a better analysis of the situations in which the retrieval does not perform well would be necessary in order to evaluate its capabilities." We agree that such an analysis would be beneficial. Therefore, we included the above mentioned new Section 4.1 in which we present further investigations of the scene structure. Besides evaluating the performance of MIBase in dependence on  $\Delta z$  and  $z_{top}$ , we also exploited how the configuration of the stereo-derived cloud mask influences the performance. This way we assessed for which scenes the algorithm performs better or worse.

"Also some statistics that quantify, in how many cases the algorithm could not retrieve a cloud base height is missing. These values should be given for each possible retrieval rejection, a too homogeneous cloud cover for instance, in comparison to the number that would have theoretically been possible."

To elaborate on this in more detail, we added Section 3.4 "Scene limitations" to the manuscript. Statistics on the situations for which MIBase cannot retrieve  $z_{\text{base}}$  are discussed quantitatively. Following the numbers in the new Table 5 and the description in the text, we now allow the reader to comprehend how we ended up with the number of cases which are considered for the calibration and validation of the algorithm. Furthermore, we also extended Section 5.1 by a discussion regarding the number of valid retrievals versus retrieval failure. Figure 10 of the revised manuscript shows the spatial distribution of scenes for which MIBase cannot retrieve  $z_{\text{base}}$ , i.e. apparent clear sky and apparent overcast.

"In Fig. 9 b), the ITCZ should be more visible in the Atlantic Ocean and over Africa, there are almost no  $z_{top}$  values over 1.4 km. Even if the analysis is restricted to cases with  $z_{top} < 5000$  m, I would assume that there should be more  $z_{tops}$  higher than 1.4 km. Could you please comment on that?"

In Fig. 9b and Fig. 10a, 10b (Fig. 11a, 11b in the revised manuscript), the ITCZ is revealed by the light turquoise band slightly north of the equator, indicating higher  $z_{\text{base}}$  and  $z_{\text{top}}$  compared to the immediate surroundings to the north and south. This band is most pronounced in the Pacific ocean. Over the Atlantic, it can be seen most

СЗ

clearly in the manuscript's Fig. 9c, which shows a band of increased cloud vertical extent in that region. As stated in the manuscript, over continents the diurnal cycle should be kept in mind. MISR has a morning overpass which means, the three year median heights provided here represent the morning heights around 10 a.m. local time. For the Congo Basin, Taylor et al. (2007) investigated the diurnal cycle of cloud top temperature (CTT) retrieved via satellite remote sensing (SEVIRI). According to them, the CTT is lowest around the MISR overpass time with a mean value of about 290 K during late morning hours. If we take the observed  $z_{top}$  of about 1200 m and assume a lapse rate of  $0.6 \frac{K}{100m}$ , the extrapolated surface temperature would be 297 K ( $\approx 24^{\circ}$ C) which seems very plausible.

**"And why is *z*\_top restricted to 5000 m, is this threshold not only applied to *z*\_base in order to exclude cirrus?"**

We agree, that the limit for  $z_{top}$  should not be the same as for  $z_{base}$ . Therefore, we reproduced the figures for the global distribution of  $z_{top}$ . This time, the median is calculated only for those  $z_{top}$  values for which the respective  $z_{base}$  is below the 5000 m threshold. We updated Fig. 9 and Fig. 11 of the revised manuscript and their respective captions accordingly. Generally, a threshold is necessary to exclude high clouds from the analysis in order to avoid difficulties associated with cirrus clouds. In our opinion the median of  $z_{base}$  and  $z_{top}$  provides less valuable information if low and high clouds are mixed together. From our best judgement, 5000 m seems like a good choice for a threshold to ensure that the algorithm works properly. The resulting product is not highly sensitive to this threshold as can be seen in Fig. 4 (supplement).

"Fig. 9 a): Since the number of valid retrievals over the Sahara is so small, it is quite understandable, that the cloud base height jumps between very small and very high values and a warning is given by the authors on page 17. In order to use maps of this kind for a climate model evaluation, many more valid data points would be necessary.

**This should be noted in the conclusions."**

We included a note of this in the conclusion: "This makes MIBase a promising tool for the evaluation of climate models on seasonal and inter-annual time scales in data sparse regions if for example the climate model output is limited to clouds below 5 km and cloud fractions below 1 and if a sufficient amount of MIBase retrievals is provided within the considered region and time period."

**"Fig. 9c): Why is the sample size low over Antarctica? Shouldn't it be covered with approx. 50% cloud cover throughout the year?"**

MISR's stereo-derived cloud mask shows configurations which indicate apparent clear sky conditions in Antarctica for 60% to almost 100% of the cases (Fig. 10c of the revised manuscript, and Fig. 3a of the supplement). This is in agreement with the cloud cover derived from MODIS presented by Suen et al. (2014).

**"p 4, I 19: please specify "SDCM" in H\_SDCM"**

SDCM stands for "stereo-derived cloud mask". We added the abbreviation in parenthesis at the first occurrence of this phrase. In particular,  $H_{\text{SDCM}}$  is the threshold height which is applied to derive the stereo derived cloud mask according to Equation 59 in the Algorithm Theoretical Basis documentation by Mueller et al. (2013). We added that this is a threshold height to the manuscript.

"p 16, I 6: "yielding an overall higher" something is missing here."

This should be "yielding an overall higher  $\Delta z$ ". Thank you for pointing this out.

"P 17, I 1: Do you refer to Fig. 9 c) instead of b)?"

Yes. Thank you and sorry for the confusion!

Please also note the supplement to this comment: https://www.atmos-meas-tech-discuss.net/amt-2018-317/amt-2018-317-AC4supplement.pdf

---

## Author Comment (AC5) · 1 Feb 2019

"The authors describe a novel algorithm for the retrieval of cloud base height (CBH) from MISR satellite measurements. Global information on cloud base height is important for many applications and the retrieval approach is interesting and promising. However, the manuscript is not sufficiently convincing in demonstrating the reliability of the new CBH product. Below are a number of major issues to be addressed before this manuscript may be suitable for publication."

We thank the reviewer for the constructive feedback. We have implemented your comments in the following way:

"The algorithm is tuned with METAR observations over the U.S., i.e. extratropical land surface. How representative is this for ocean surfaces and other climate zones, where different cloud types prevail? To show the skill in other regions, some comparisons with independent measurements elsewhere would be required."

Our study includes cloud height retrievals over the continental U.S. over the course of two years (one year for calibration, one for validation). Therefore, various cloud types should be included in the analysis. Clouds within Arctic air masses which typically occur in the northern U.S. during winter, as well as tropical like deep convective clouds which typically occur during the summer in the southeastern U.S. should be included as well as stratocumulus clouds which usually occur at the coast of California. The METAR data set includes maritime island stations in the Gulf of Mexico and near the west coast of the U.S.

The utilized MISR product does not distinguish between land and ocean, but only between cloud and surface by a geometric technique. Furthermore, for each retrieval scene a particular configuration of the stereo-derived cloud mask is provided which characterizes the scene structure. Therefore, if a similar scene structure is found outside the continental U.S. region, for which the calibration and validation has been carried out, MIBase should perform similarly. However, we agree, that additional validation in other regions would be beneficial to backup this statement.

To further investigate MIBase limitations in respect to cloud types, we introduced two new sections "3.4 Scene limitations" and "4.1 Scene structure influence" into the manuscript and modified abstract and conclusions accordingly. In short: The statistics are rather robust with regard to the configuration of the stereo-derived cloud mask. The bias depends on the number of z retrievals marked high confidence cloud. This indicates that a bias correction might be feasible. Since the origin of the bias is not fully understood yet, we like to leave such potential improvements to future studies.

Furthermore, the new Section 4.1 includes an investigation of the influence of  $z_{top}$
and  $\Delta z$  on the MIBase performance. Such parameters are also important for cloud type classification. High correlation coefficients (supplement Fig. 2h) and low RMSE (supplement Fig. 2f) for cloud thicknesses up to 1000 m indicate that the algorithm works particularly well for thinner clouds. For cirrus clouds or high clouds in general, we cannot make a robust statement as we tax the accuracy of the METAR reports insufficient for this particular height range.

"More information on the success rate of the retrieval algorithm is required to evaluate how useful it is. Statistics of the number of samples n\_s are given but these are only absolute numbers, not (fractional) success rates. For example, Table 3 indicates that n\_s is between 3059 to 7772 depending on R\_fv. A rough calculation based on 1510 ceilometers, a MISR revisit time of 6 days, and a cloud fraction of 50% would potentially yield around 45,000 cloudy collocations. This suggests that in only 10% of the cloudy cases, a valid CBH retrieval is obtained. Is this correct? Such statistics, accompanied by the relative occurrence of different causes of retrieval failure, need to be provided, also for the global plots, to evaluate the applicability of the method."

We added statistics regarding the success rate of the algorithm to the manuscript (Section 3.4 and Section 5.1 of the revised manuscript). For the comparison with the retrievals from METAR, we combined the numbers for all sites and present the resulting numbers for the years 2008 (calibration) and 2007 (validation) individually in the new Section 3.4 including new Table 5. For the year 2008, we downloaded data which provided 80454 overpasses. Only 65% of those contained valid *z* retrievals at the METAR sites with a corresponding METAR message. Out of those potential cases, about 30% do not include *z* retrievals marked high confidence surface. Please, see the revised manuscript for more details. For the global application, the calculations are carried out for each grid cell, so that the spatial distribution of the numbers can be studied (Fig. 10 of the revised manuscript). We added a discussion on the retrieval failure statistics to Section 5.1.
"The calibration of the algorithm is done for *z*\_base smaller than 3000m, because of the limited range of the ceilometers. However, for the global composites an upper threshold of 5000m is used. It is unclear whether this extrapolation outside the range for which the product has been trained, is valid."

One of the questions this manuscript tries to answer is whether the cloud base height can be derived from the MISR cloud top height product. The comparison with the ceilometer demonstrates that this is possible in case some preconditions are met. The reason this works is because typically the cloud top height is heterogeneous leading to geometrically thinner and thicker parts of the cloud. As far as we understand, this concept should not change for different heights within the troposphere. This means, if it can be validated within one height region, it should work in other height regions as well. However, we do agree that the algorithm might perform differently for different cloud types. Then of course, for different heights the distribution of cloud types varies. We discussed the dependence on cloud types in the reply to the first comment.

"The global maps in Figs. 9 and 10 are hard to interpret because upper limits of *z*\_base and *z*\_top have been applied. What does the median of a distribution cut-off at some value tell us?"

Applying a threshold is necessary to exclude high clouds from the analysis. This is appropriate, because we are focusing the study on clouds which occur in the lower troposphere. In our opinion, the median of  $z_{base}$  and  $z_{top}$  provides less valuable information if low and high clouds are mixed together. From our best judgement, 5000 m seems like a good choice for a threshold to ensure that the algorithm works properly. The resulting product is not highly sensitive to this threshold as can be seen in Fig. 4 (supplement). Inspired by reviewer 6 (RC4) who questioned the height limit for  $z_{top}$ , we changed the calculation of the median  $z_{top}$  height. We reproduced the figures by calculating the median only for those  $z_{top}$  values for which the respective  $z_{base}$  is below the 5000 m threshold. We updated Fig. 9 and Fig. 11 of the revised
manuscript and their respective captions accordingly.

"I'm also confused by the description of Fig. 9, which says that the ITCZ is clearly visible with higher *z*\_base and *z*\_top. In the plots a brown band can be seen, but these are lower rather than higher values. Can you explain?"

In Fig. 9b and Fig. 10a, 10b (Fig. 11a, 11b in the revised manuscript), the ITCZ is revealed by the light turquoise band slightly north of the equator, indicating higher  $z_{\text{base}}$  and  $z_{\text{top}}$  compared to the immediate surroundings to the north and south. This band is most pronounced in the Pacific ocean. Over the Atlantic it can be seen most clearly in the manuscript's Fig. 9c, which shows a band of increased cloud vertical extent in that region. As stated in the manuscript, over continents the diurnal cycle should be kept in mind. MISR has a morning overpass which means, the three year median heights provided here represent the morning heights around 10 a.m. local time. For the Congo Basin, Taylor et al. (2007) investigated the diurnal cycle of cloud top temperature (CTT) retrieved via satellite remote sensing (SEVIRI). According to them, the CTT is lowest around the MISR overpass time with a mean value of about 290 K during late morning hours. If we take the observed  $z_{\text{top}}$  of about 1200 m and assume a lapse rate of  $0.6 \frac{K}{100\text{m}}$ , the extrapolated surface temperature would be 297 K ( $\approx 24^{\circ}$ C) which seems very plausible.

"Is it also possible that the results in these multi-year median are biased to certain cloud types? For example, in the stratocumulus (Sc) areas west of the continents, cases with closed Sc will probably not yield a valid retrieval, while for open Sc z\_base can be retrieved, so that the end result will be biased to the latter."

An inherent bias of the method results by the necessary condition of a MISR z retrieval which is marked high confidence surface by the stereo-derived cloud mask. In other words, a broken cloud scene is required. Therefore, cloud base heights for situations
with overcast are not included in the calculation of the three-year median. So yes, the statistic is biased towards particular cloud types. We stated this limitation more clearly in the manuscript by mentioning the limitation in Section 3.1 and additionally in the conclusion. Apparent overcast situations which prevent a valid MIBase retrieval occur mainly in the mid latitudes over ocean and in the subtropical stratocumulus areas which you mentioned as an example (Fig. 10d of the revised manuscript).

"The authors define percentile (P) values of the MISR lowest cloud layer z distribution to obtain z\_base and z\_top. For z\_base one would expect P=0 because z\_base should be lower than any MISR-derived cloud-top height. The chosen value P=15 is motivated by the noise in MISR z, which makes sense. However, for z\_top I do not understand the chosen value P=95. All MISR z values are actual estimated cloud top heights. The logical way to aggregate these is to average the individual z observations or take the median. In other words, a value P=50 seems natural. The choice of P=95 should thus be motivated."

Without further validation, we apply the 95th percentile rather than the median, as we do not want a height which might be representative for the whole area, but rather an estimate of the highest top of the cloud especially for a heterogeneous cloud top height. The focus of this study is on the  $z_{base}$  retrieval method and its validation. The use of  $z_{top}$  retrievals serves only auxiliary purposes. It is a measure to describe the individual cloud scenes better. For instance, it allows a qualitative assessment of the algorithm's performance in dependence on cloud vertical extent. We extended the motivation of the 95th percentile at the end of Section 3.1.

"Cloudsat is, especially in combination with Calipso, arguably the most accurate source of cloud base height (as well as cloud top height) information from space. Surprisingly, Cloudsat is not mentioned at all (except for two remarks in the context of the Desmons et al. paper) in the manuscript. At the least, Cloudsat should be discussed, and it
would also be good to make some comparisons with this instrument, even if direct collocations with MISR supposedly only occur at high latitudes."

We agree, that it would be very beneficial to have further data sets to compare the method to. However, CloudSat has limitations in estimating the cloud base height, in particular for low liquid clouds. It does not detect small droplets at the base of the cloud (Sassen and Wang, 2008) due to its detection limit of  $\approx -28 \, \text{dBz}$ . Furthermore, retrievals are degraded in the ground clutter reigon (Tanelli et al., 2008; Marchand et al., 2008). Mülmenstädt et al. (2018) evaluate the 2B-GEOPROF-LIDAR product (Mace and Zhang, 2014) which uses a combination of CloudSat and CALIOP retrievals. From this product, they extracted the LIDAR only and RADAR only subsets and compared the cloud base height retrievals with ceilometer measurements similar to the reference data utilized in this study (their Fig. 9). Within their study, the RADAR does not perform as well as the LIDAR. In fact, they find a correlation of 0.265 and an RMSE of 782 m for the RADAR only subset. Therefore, we believe CloudSat would not be suitable as reference data for our study.

A comparison with Cloudsat would require the identification of collocated measurements, which you mentioned would occur at high latitudes. CloudSat has an afternoon overpass, while MISR on Terra has a morning overpass. Therefore, a comparison would also require a discussion on the impact of the temporal difference. We believe, that this paper should focus on the introduction of this new  $z_{base}$  retrieval method, its calibration and validation, statistics on the success rate, and a discussion on its global application. Further comparisons go beyond the scope of this paper.

"P1, Abstract: The abstract does not include any information on the cloud types the algorithm is applicable to. In the manuscript this information is also too limited. Does the method work for cirrus, or for deep convective clouds?"

We added that overcast cloud scenes are not included in the statistics to the abstract.
Further, we added "The impacts of the cloud scene structure and macrophysical cloud properties discussed", to alert the reader, that this is an issue.

Our study does not focus on cirrus clouds. Since the MISR *z* retrieval method is in the visible light range, thin cirrus clouds are probably not included, since they are almost transparent. Further, we cannot validate cirrus clouds, as the METAR data does not include reliable retrievals for high clouds. Deep convection might be problematic, as the thinner cloud edge might not be as pronounced or hidden by the towering cloud. However, deep convection should not be a major issue, even in tropical regions, because of the morning overpass time of MISR on Terra.

"P2, L8: Stephens et al. (2002) is mainly about Cloudsat. It's not the appropriate reference for CALIPSO."

Thank you for pointing this out. We changed the reference by citing Winker et al., 2010.

"P4, L4: Is 'aftward' correct English?"

We think aftward is correct English. It is also used in the MISR product documentation. However, we will follow whatever guideline the editor suggests.

"P5, L6: 'measurements': what is measured?"

The signal return is measured. We rephrased the sentence.

*"P5, L9: Is the value 5000 ft correct? It seems such a big jump from 100 ft and 200 ft in the two lower height categories, respectively."*

Sorry, that is a typo. According to the "Automated Surface Observing System User's Guide" (National Oceanic and Atmospheric Administration, Department of Defense, Federal Aviation Administration, and United States Navy, 1998) it is  $500 \text{ ft} (\approx 150 \text{ m})$ .

AMTD
"P5, L10: This suggests that bins is the same as clusters, which is not the case, I assume."

We delete "bins" from the sentence.

"P8, L1-3: This paragraph looks out of place here. Suggest to move it somewhere else."

We moved it to the end of Section 2.1.

"P8, L7: Suggest to replace 'the estimated' by 'a typical'."

Done.

"P8, L15: what are 'z pixels'?"

It refers to the z retrievals from the MISR cloud product. We replaced "pixel" by "retrieval".

"P8, L25-26: Does this mean that the case in Fig. 4 is not included in the statistics? Isn't it a bit strange to present a case study that is not part of the selection applied furtheron?"

This case was studied before we decided which years we would use for the comparison. The main reason why it is shown, is because it illustrates the way the algorithm works and the parameters which are used. Preferentially, the presented case study should be a multi-layer case so the applied layer distinction can be illustrated as well. However, as mentioned in the manuscript, any multi-layer case will be masked out and not be included in the selection for further processing in order to calibrate the algorithm. AMTD
"P10, L5: Fig. 4 includes only one 'h\_gap'."

It is only illustrated once. However,  $h_{gap}$  is calculated for any height gap between two *z* retrievals and then tested against the threshold. If it is greater than 500 m, the retrievals below and above the gap are treated as separate layers. In the case study, an apparent third layer which is about 1000 m above the top of the middle layer, is revealed in the density plot (Fig. 4 in the manuscript).

"P10, L13-14: The second layer detected by MISR has a base height between 5000 and 5500 m a.s.l. The ceilometer detects layer base heights at 853 m, 2286 m, and 7010 m a.g.l. None of these seems to match with the second MISR layer. Can you explain?"

Thank you for pointing this out. From the distribution of the *z* retrievals (Fig. 4 in the manuscript), we can distinguish three apparent cloud layers. The highest ceilometer retrieval seems to correspond well with the top layer (between 7000 m and 8000 m). The lowest ceilometer retrieval corresponds well with the bottom layer. We corrected the last sentence of Section 3.2 accordingly. The second ceilometer retrieval roughly matches the top of the lowest layer detected by MISR. The connection to the bottom layer detected by MISR might be an indication of a varying cloud base height within this cloud field. It could also be due to the temporal mismatch between the measurements.

"P12, Fig. 6: I assume this figure is for N=10 and R\_fv=10 km. Can you add this to the caption for completeness?"

Yes. Done.

"P12, L6: N=10 seems a relatively low number and one could wonder whether P=15
**makes sense for such a small N. Can you comment?"**

As discussed in the manuscript, N = 10 is a compromise. If a higher N is chosen, the performance improves only slightly. At the same time, the algorithm would neglect more potential retrieval scenes. As addressed above, the bias increases for an increasing number of z retrievals marked high confidence cloud  $N_{\text{HCC}}$  (Fig. 8k of the revised manuscript). This indicates the potential for a bias correction. Another way to decrease the bias could be carried out by defining the selected percentile as a function of  $N_{\text{HCC}}$ . However, as mentioned above, the origin of the bias is not fully understood yet. Therefore, we like to leave such potential improvements to future studies.

**"P13, Fig. 7: Can you add a color bar? Is the scale linear or logarithmic?"**

The color indicates a normalized density. It is a linear scale. Contour lines are shown with the corresponding values on them. We modified the caption to point this out.

"P14, Table 4: Do you have any explanation why 2007 has 30% more valid retrievals than 2008?"

We found about 18% more overpasses with valid z retrievals in the fields around the METAR sites. And out of those we did not have to neglect as many apparent clear sky cases. See the new section 3.4 in the manuscript for more details.

"P14, L5: Certainly the different measurement geometry (point over time versus circular area instantaneous) can cause differences. But why would this lead to a bias, and why to a negative bias of MISR in particular? Can't you tune the overall bias to near zero by increasing P?"

As we added to the manuscript, "the bias obtained in this study can have different sources: the different sample volumes of the defined MIBase cell and the ceilometer,

AMTD
biased MISR *z* retrievals, various scene characteristics." As of now, we found that the bias seems to depend strongly on  $N_{\rm HCC}$ . Simply modifying *P* to tune the bias to zero is overlooking this relationship. As mentioned earlier, an adaptive *P* depending on  $N_{\rm HCC}$  or an appropriate bias correction would improve the algorithm. However, this goes beyond the scope of this study.

*"P15, Fig. 8: I'm not sure how useful this distinction in two geometrical thickness classes is, in particular because this thickness is based upon the MISR retrieval itself."*

We substituted this distinction by a more in depth discussion about the influence of the scene structure (new Section 4.1). The usage of  $z_{top}$  and  $\Delta z$  provides additional information which characterizes the scene structure beyond just the value of  $z_{base}$ . Therefore, it is justified, to use this information to further study the performance of the algorithm.

"P15, L3: The termination of the *z*\_base range by the threshold height relates mostly to the lower thickness class, so it would be better to write: 'The smaller E for clouds with a smaller Delta *z* ..'."

We cut this part out.

"P16, L6: Sentence ends unexpectedly."

Fixed. Thank you.

*"P17, L1: The sampling size is in Fig. 9c."* Yes. Fixed. Interactive comment

"P19, L3: effect should be affect."

**Fixed**

"P19, L32: A mean difference of 500 m is quite large relative to the retrieved  $z_base$  in Fig. 11, which appears to vary between 800 and 1200 m for the selected region. Is it reasonable to assume that the model can simulate a reasonable seasonal variation of  $z_base$  if it has such a large bias?"

Yes, we think, it is reasonable. The processes responsible for defining the height of the cloud base are different from the processes which produce the seasonal cycle. Models generally underestimate the maritime boundary layer height in the stratocumulus regions. However, the radiation forcing, and the strength of the subsidence which follow an annual cycle can be represented in the model with higher accuracy leading to a realistic seasonal cycle, despite the revealed bias.

Please also note the supplement to this comment: https://www.atmos-meas-tech-discuss.net/amt-2018-317/amt-2018-317-AC5supplement.pdf

---

## Author Comment (AC6) · 1 Feb 2019

*"As there are already five reviews available (and more referees have accepted the review of the paper) I can be short with my statements. I agree with what has been mentioned by the other reviewers with respect to the pre-conditions (cloud optical thickness, homogeneity), so I can restrict myself to comments mainly related to the ceilometersas this has not yet been covered in detail."*

Thank you for your constructive feedback. We have implemented your comments as explained below.

*"Section 2.1: The expression "z pixel" might be revised/improved"*

[Figure]

We replaced "$z$ pixel" by "$z$ retrieval".

*"Section 2.2: Please add 1-2 sentences to describe the type of ceilometers used, and the basic characteristics of the instrument and the cloud height retrieval."*

We added that the ceilometers are lidar ceilometers which are operating at a wavelength of $0.9\,\mu$m to Section 2.2. Additionally, we mention that the cloud base height retrievals are derived by evaluating the vertical gradient of the backscatter profile.

*"Is the very coarse vertical resolution of the METAR-messages an issue?"*

Due to the rounding, the given vertical resolution of the METAR $\hat{z}_{\mathrm{base}}$ reports is 100 ft ($\approx 30\,$m) for heights up to 5000 ft ($\approx 1500\,$m) and 500 ft ($\approx 150\,$m) between 5000 ft and 10000 ft ($\approx 3000\,$m). We expect the uncertainty of the MIBase retrievals to be larger than this (RMSE $\approx 400\,$m), so that the resolution of the METAR messages is a small contribution to the total uncertainty.

*"What about using backscatter profiles from ceilometer networks, e.g. in Europe: derived cloud base heights are quite reliable and the vertical resolution is in the order of 10 meters. Please comment on this; maybe in the conclusions."*

Due to the large homogeneous data set, we focus on the continental U.S. We are aware of harmonisation efforts within Europe. Therefore, we added the following sentence to the conclusion: "Within Europe, the European Cooperation in Science and Technology (COST) activity is expected to harmonize the networks of the different weather services (e.g. Haeffelin et al., 2016 and Illingworth et al., 2018, for further reading) enabling more inter comparisons in the future."

*"Is the variability of the 30 s messages used to exclude certain data sets (temporal variability translated to spatial inhomogeneity [taking into account the bins of the messages])?"*

As far as we know from the ASOS handbook, no filtering for inhomogeneity is performed.

*"The discussion of the implications of the time period of 30 minutes for averaging could be extended."*

We added the following sentences to the new Section 3.4: "The METAR reports comprise retrievals over a 30 minute period. During this time, cloud formation and cloud dissipation can alter the cloud scene and cause mismatches between MISR and METAR retrievals."

*"Section 3: Better use another word for "field of view" ($R_f$) here: according to page 7, line 11 it has nothing to do with the optics of the radiometers onboard of MISR as one might expect."*

We agree that "field of view" is inappropriate here. We changed it to "MIBase cell" throughout the manuscript. For consistency, we also modified the notation for the radius which defines the size of the MIBase cell from $R_{\mathrm{fv}}$ to $R_{\mathrm{c}}$.

*"Section 3.2: Taking into account the very poor vertical resolution of the ceilometers and the large "footprint" of the inter-comparison I feel that it is not justified to end up with a $\hat{z}_{\mathrm{base}} \approx 853$ m (pretending a one-meter-accuracy). Can you give an uncertainty instead of using "≈"."*

As stated above, the binning during the data processing of the ceilometer measurements, leads to a vertical resolution of the METAR retrievals between 100 ft ($\approx 30$ m) and 500 ft ($\approx 150$ m). This resolution should suffice for the analysis carried out in this study. The native METAR ceiling report was $2800$ ft which is an integer multiple of the measurement resolution. Here we convert to SI units, which leads to values which look not round at all. To avoid the illusion of one meter accuracy, we changed that particular instance to $\hat{z}_{\mathrm{base}} = (853 \pm 15)$ m and added: "Since METAR values are rounded to the nearest $100$ ft and no information on uncertainty is available, we

estimate an uncertainty of approximately $15\,\mathrm{m}$."

*"Page 10, line 17 states that a cloud base height of 7010 m was retrieved. In section 2.2 it is stated that the ceilometers have a vertical range of up to 3700 m. Please explain."*

This height was included in that particular METAR message. This can happen, because a subset of the ceilometers has a higher measurement range. In case of multiple layers, and if at least the lowest retrieval occurs within the reporting range, cloud heights outside this range can be included in the report.

*"The caption of Fig. 12 could be misleading. Mention that deviations are shown right at the beginning of the text."*

We edited the caption and the axis labels.

*"The conclusions of the papers cited in Hannay et al. (2009) are mainly based on thermodynamics. They do not cover pbl-retrievals based on backscatter. This is however relevant for ceilometers (that are used as reference in this paper). Therefore the agreement/disagreement of ceilometer-retrievals with model results should be discussed as well: a lot of papers have recently been published focussing on the potential of ceilometers in general and the determination (and its accuracy) of the mixing layer height (or pbl)."*

The reason why we are citing Hannay et al. (2009) is that they provide studies from the area we are interested in, i.e. the southeast Pacific. Their comparison to observations based on radiosonde data and microwave radiometer retrievals shows that the models underestimate the boundary layer height in this region where stratocumulus clouds prevail. Their conclusion should not be generalized outside this area. To clarify that

the study by Hannay et al. is carried out over the southeast Pacific, we updated the manuscript accordingly. We agree, that where available Lidar and ceilometer measurements would be beneficial to validate the mixing layer heights and cloud heights from models. However, we are not aware of such comparisons for this particular region.

*"I agree that the MIBase can be a promising tool for remote areas, and for climatological studies with the corresponding (extended) spatiotemporal averages. Nevertheless a few comments on the benefit of the retrieval based on individual observations would be desirable, considering the large uncertainty and the missing coverage of the diurnal cycle. So combination with ground based ceilometer networks (where available) should be envisaged, especially as ceilometers are a very direct and accurate approach (no calibration required, continuous operation) for $z_{\text{base}}$-retrievals."*

We added "Depending on the application, the MIBase uncertainty and the missing coverage of the diurnal cycle can be a limitation. However, in combination with ceilometer networks, both temporal and spatial patterns can be investigated." to the conclusion.

Please also note the supplement to this comment:
https://www.atmos-meas-tech-discuss.net/amt-2018-317/amt-2018-317-AC6-supplement.pdf

**Supplement:**

[Figure]

**Figure 1:** a) Normalised frequency of occurrence of the occurring standard deviations (sd) of the average scene elevation (ASE) for various radii. The ASE is provided by the MISR ancillary product. Recalculated are the correlation coefficient $r$, bias, RMSE using only ceilometer stations below a threshold sd as denoted on the abscissas.

[Figure]

**Figure 2:** Number of samples $n_s$, RMSE, bias, correlation coefficient $r$ for the comparison of MIBase and ceilometer retrievals in dependence on $z_{top}$ (top row) and cloud vertical extent $\Delta z$ (bottom row). Each data point is calculated for a sub sample which includes only $z_{top} \pm \delta z_{top}$ or $\Delta z \pm \delta \Delta z$, respectively. The various widths of the considered $z_{top}$ or $\Delta z$ windows are indicated by the blue shading.

[Figure]

**Figure 3:** Relative occurrences of different stereo-derived cloud mask (SDCM) configurations within the three-year period (2007-2009). The reference sample size $n_s$ includes all overpasses per grid cell which contain valid $z$ retrievals and corresponds to 100 %. These configurations are: (a) Only high confidence surface (HCS). These cases should be mainly clear sky cases. (b) Only high confidence cloud (HCC). These cases should be mainly cloud scenes with apparent overcast.

[Figure]

**Figure 4:** Global distribution of median cloud heights for a 3-year period (2007–2009). Shown are $z_{base}$ (left) and $z_{top}$ (right) on a 0.25° × 0.25° latitude–longitude grid. $z_{base}$ and $z_{top}$ are above ground level (agl). $z_{base}$ and $z_{top}$ retrievals are only included in the statistic if $z_{base}$ is below 3000 m (a, b), 5000 m (c, d). For (e) and (f), all $z_{base}$ and $z_{top}$ retrievals are included without an upper height limit.